# A Brain-Inspired Gating Mechanism Unlocks Robust Computation in Spiking Neural Networks

**Qianyi Bai**[1,2,*], **Haiteng Wang**[1,3,*], **Qiang Yu**[1,†]

[1]School of Artificial Intelligence,
Tianjin Key Laboratory of Cognitive Computing and Application,
College of Intelligence and Computing, Tianjin University, Tianjin, China
[2]School of Computer Science and Technology, Tianjin University, Tianjin, China
[3]School of Future Technology, Tianjin University, Tianjin, China
{baiqianyi, wanght_0103, yuqiang}@tju.edu.cn

## Abstract

While spiking neural networks (SNNs) provide a biologically inspired and energy-efficient computational framework, their robustness and the dynamic advantages inherent to biological neurons remain significantly underutilized owing to over-simplified neuron models. In particular, conventional leaky integrate-and-fire (LIF) neurons often omit the dynamic conductance mechanisms inherent in biological neurons, thereby limiting their capacity to cope with noise and temporal variability. In this work, we revisit dynamic conductance from a functional perspective and uncover its intrinsic role as a bio-inspired gating mechanism that modulates information flow. Building on this insight, we introduce the Dynamic Gated Neuron (DGN), a novel spiking unit in which membrane conductance evolves in response to neuronal activity, enabling selective input filtering and adaptive noise suppression. We provide a theoretical analysis showing that DGN possess enhanced stochastic stability compared to standard LIF models, with dynamic conductance intriguingly acting as a disturbance rejection mechanism. DGN-based SNNs demonstrate superior performance across extensive evaluations on anti-noise tasks and temporal-related benchmarks such as TIDIGITS and SHD, consistently exhibiting excellent robustness. To the best of our knowledge, for the first time, our results establish bio-inspired dynamic gating as a key mechanism for robust spike-based computation, providing not only theoretical guarantees but also strong empirical validations. This work thus paves the way for more resilient, efficient, and biologically inspired spiking neural networks.

## 1 Introduction

Spiking Neural Networks (SNNs) offer a biologically inspired alternative to traditional neural architectures by leveraging discrete, event-driven spikes for computation. Their energy efficiency, temporal expressiveness, and robustness to noise make them increasingly attractive for neuromorphic applications (Maass, 1997; Pfeiffer & Pfeil, 2018; Roy et al., 2019; He et al., 2020; Yu et al., 2025). As third-generation networks, SNNs aim to bridge the gap between artificial computation and biological realism (Pei et al., 2019). However, despite their biological motivations, most existing SNN models—commonly referred to as Gateless SNNs—lack internal gating mechanisms for modulating neuronal dynamics (Bellec et al., 2018; Fang et al., 2021; Zhang et al., 2024a). Recent efforts, such as the Gated LIF (GLIF) model (Yao et al., 2022), introduce static, channel-wise gates but remain biologically implausible. As a result, the field still lacks a biologically grounded dynamic gating mechanism for spiking neurons—a fundamental obstacle to developing more adaptive and robust SNNs.

---

[*]These authors contributed equally. Pursuing a master's degree under the supervision of Prof. Q. Yu
[†]Corresponding author.

Protein phosphorylation and gene expression have demonstrated that ion channel conductance in biological neurons are not static; they can be dynamically modulated in response to sustained neural activity (Kaczmarek, 1987; Chad & Eckert, 1986; Morgan & Curran, 1991; Sheng & Greenberg, 1990). For instance, depolarization can trigger the expression of immediate early genes such as fos and ras, leading to changes in potassium conductance (Smeyne et al., 1992), while prolonged depolarization has been observed to reduce calcium currents (Sharp et al., 1993). Intracellular calcium often mediates these processes, serving as a second messenger that links neuronal activity to conductance modulation (Kaczmarek & Levitan, 1987). These findings have inspired computational models that incorporate dynamic regulation of membrane conductance. Beyond their role in homeostatic adaptation, such conductance modulations can be viewed as intrinsic gating mechanisms, wherein membrane properties dynamically shape neuronal responsiveness based on prior activity. This biologically grounded form of gating operates independently of synaptic transmission and plays a central role in regulating neural computation.

Building on a series of biologically grounded studies on conductance-based neurons—including the influential work by Gütig (Gütig & Sompolinsky, 2009), which formulated their dynamic equations and revealed their time-warp–invariant property—we revisit this class of models to bridge the gap between biologically inspired dynamics and their underexplored integration into spiking neural network frameworks. We reintroduced the dynamic conductance mechanism into the LIF neuron model and proposed the Dynamic Gated Neuron (DGN) model. In DGN model, membrane conductance are dynamically modulated as a function of incoming activity. This process implements a bio-inspired gating mechanism that adaptively modulates the persistence of internal states based on input dynamics, allowing spiking neurons to control the retention and decay of past information—functionally analogous to gating operations in recurrent architectures such as the forget gate in LSTMs (Hochreiter & Schmidhuber, 1997). While models like LSTMs and GRUs (Cho et al., 2014) have achieved remarkable performance through engineered gating schemes, their designs are largely disconnected from biological mechanisms. By grounding gating dynamics in neurophysiological principles, our approach bridges this gap, offering a unifying theoretical framework that links spiking neural models with artificial gated recurrent units. This biologically inspired perspective not only enhances the interpretability of gating functions in artificial systems, but also promotes the development of more robust and adaptive architectures informed by the dynamics of real neural circuits.

Unlike traditional LIF neurons, which simplify neural dynamics by using fixed decay rates and static conductance parameters, our model introduces input-dependent modulation of membrane conductance. This enables neurons to selectively retain relevant information while suppressing irrelevant or noisy inputs, thereby implementing a bio-inspired gating mechanism. We evaluate the proposed model within multi-layer spiking neural networks and it achieves strong classification performance while demonstrating stronger resistance to noise and perturbation. Our contributions are summarized as follows:

- **DGN:** We propose the Dynamic Gated Neuron (DGN) model, a generalized spiking neuron model with a fully derived membrane potential formulation. Central to DGN is a dynamic conductance mechanism that functions as a bio-inspired gating mechanism, enabling adaptive control over information flow and memory retention within the neuron. DGN underscores the critical role of dynamic conductance, a biological mechanism, in the modeling of neuronal dynamics.

- **Bridging Biologically Inspired Dynamics and Artificial Gating Mechanisms:** We identify functional parallels between dynamic conductance modulation in our model and gating mechanisms in LSTM networks, offering a biologically grounded perspective that helps bridge the gap between brain-inspired computation and artificial neural networks.

- **Robustness Analysis and Accuracy Results:** We present a complete theoretical analysis of the anti-perturbation properties arising from dynamic conductance mechanisms. In addition, we conduct anti-noise experiments on benchmark datasets using the DGN model, which consistently demonstrates strong performance across both audio and neuromorphic tasks. Notably, our model achieves state-of-the-art top-1 accuracy of 99.10% on the TIDIGITS dataset using a small network.

## 2 RELATED WORK

### 2.1 BIOLOGICAL AND COMPUTATIONAL PARAMETRIC NEURON MODELS

The Hodgkin-Huxley (HH) model (Hodgkin & Huxley, 1952) introduced a biophysically detailed, conductance-based description of neuronal dynamics, capturing action potential generation via voltage-gated ion channels. Despite its accuracy, the computational cost of solving HH equations limited its adoption in spiking neural networks (SNNs), which typically rely on oversimplified models such as LIF (Koch & Segev, 1998) and SRM (Gerstner et al., 2014). Izhikevich's comparative analysis (Izhikevich, 2004) further highlighted the trade-offs between biological plausibility and computational efficiency in neuronal modeling. While conductance-based models offer richer dynamic properties, their integration into SNNs remains rare. In contrast, our work leverages dynamic conductance not merely for biophysical fidelity, but as a functional gating mechanism, bridging biological modeling with modern computational frameworks. To enhance the temporal modeling capacity of spiking neurons, recent studies have extended the classical LIF framework by incorporating more flexible parameterizations, leading to the emergence of Computational Parametric Spiking Neurons (Yao et al., 2022). Representative models include the Adaptive LIF (ALIF) neuron (Bellec et al., 2018), which introduces activity-dependent threshold adaptation; GLIF (Yao et al., 2022), which embeds gating mechanisms to modulate membrane potential dynamics; Heterogeneous LIF (Perez-Nieves et al., 2021), which enables learnable membrane time constants; and FS-neuron (Stöckl & Maass, 2021), which treats all membrane-related parameters as trainable. A recent model introduces a double-threshold mechanism to enable both positive and negative spike generation (Zhou et al., 2024). HetSyn (Deng et al., 2025) shifts temporal integration from the membrane potential to the synaptic current. While these models improve expressiveness through structural extensions or trainability, our approach is more biologically grounded: it incorporates dynamic conductance as a functional gating mechanism, enabling adaptive regulation of information flow and memory retention with competitive performance across tasks.

### 2.2 ROBUSTNESS ON SNNS

To enhance the robustness of SNNs against noise and adversarial perturbations, prior works can be broadly categorized into three types: structural modeling, training-based strategies, and biologically-inspired mechanisms. Structural approaches focus on neuron-level properties; for instance, adjusting firing thresholds and temporal windows significantly affects adversarial robustness (El-Allami et al., 2021), and precise spike timing has been shown to stabilize temporal representations (Ding et al., 2023). Other works have investigated the role of membrane potential leakage in LIF neurons, demonstrating that proper tuning can suppress high-frequency perturbations (Sharmin et al., 2020; Chowdhury et al., 2021). Recent methods further propose learnable and heterogeneous leak factors to adaptively regulate information retention across time steps (Fang et al., 2021; Perez-Nieves et al., 2021; Ding et al., 2024a). Training-based methods improve robustness by injecting adversarial examples during learning (Kundu et al., 2021) or applying Lipschitz regularization to limit gradient sensitivity (Ding et al., 2022), but they typically rely on static input encoding and overlook temporal dynamics. Biologically inspired strategies mimic mechanisms observed in neural systems, such as introducing stochastic gating to emulate biological randomness (Ding et al., 2024b), or leveraging frequency-based encoding to simulate selective attention by filtering high-frequency components at different time steps (Xu et al., 2024). Together, these efforts highlight the importance of combining robust training objectives with biologically-aligned temporal regulation to improve SNN robustness. Building on these insights, our DGN model proposes a novel robustness-enhancing framework that more closely aligns with biological neural dynamics, which is our dynamic conductance scheme.

## 3 METHODOLOGY

### 3.1 DYNAMIC GATED NEURON MODEL

Extensive research on neuronal conductance mechanisms has established diverse conductance-based models (FitzHugh, 1961; Morris & Lecar, 1981; Hille, 2001; Wilson & Cowan, 1972; Gütig & Sompolinsky, 2009), enhancing biological plausibility beyond traditional LIF frameworks. The

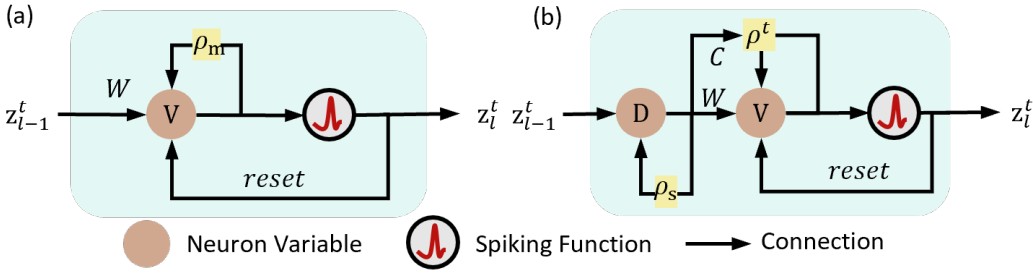

Figure 1: Schematic of the neuron models. (a) a standard LIF model ($\rho_{\mathrm{m}} = e^{-g_l \Delta t}$). (b) the as-proposed DGN model described in Eq. 5-8 ($\rho_{\mathrm{s}} = e^{-\frac{\Delta t}{\tau_s}}$).

neuronal dynamics of a basic conductance-based neuron can be described by the following formula:

$$\frac{dV}{dt} = -g_l V + \sum_i^N g_i(E_i - V) \tag{1}$$

$$\frac{dg_i}{dt} = -\frac{1}{\tau_{\mathrm{s}}} g_i + C_i \sum_j \delta(t - t_i^j) \tag{2}$$

where $g_l$ is the leak conductance. $g_i$ represents the conductance of $i$-th synapse. $N$ signifies the number of presynaptic afferent. $t_i^j$ indicating the arrival time of the $j$-th presynaptic spike of the $i$-th afferent neuron before time $t$. $\tau_{\mathrm{s}}$ is the synaptic time constant. $C_i$ represents learning weights of conductance. $E_i$ represents the equilibrium potential of the $i$-th synapse. $E_i$ has excitatory synaptic values and inhibitory synaptic values.

Further analysis of conductance-based neuron models revealed that membrane conductance ($g_l + \sum g_i E_i$) exhibits activity-dependent plasticity modulated by presynaptic spiking patterns. This synaptic-driven mechanisms precisely regulate the decay rate of membrane potential, thereby modulating neuronal memory efficiency and temporal integration properties. Inspired by this, we present a Dynamic Gated Neuron (DGN) model that implements a biologically grounded gating structure to reconcile neuronal biophysics with computational efficiency. This framework introduces dynamic conductance as a fundamental gating mechanism, emulating biological neurons' adaptive signal integration while preserving critical information retention properties. The model's mathematical formulation controls membrane potential dynamics through two interacting gating components: input-dependent synaptic conductance and intrinsic leak conductance. The temporal evolution of membrane potential $V$ obeys the differential equation:

$$\tau_{\mathrm{s}} \frac{dD_i}{dt} = -D_i + z_i^t \tag{3}$$

$$\frac{dV}{dt} = -(g_l + \sum_i^N C_i D_i)V + \sum_i^N W_i D_i \tag{4}$$

where $z_i^t$ is the input spike of the $i$-th synapse in time $t$. $D_i$ is the exponentially decaying synaptic current to soma of the $i$-th synapse. $W_i$ is learning weights of input current. Detailed derivations of these neuronal dynamics are provided in Appendix A.1.1. For practical implementation of SNNs based on connected spiking neurons, coupled with spike firing and spike resetting processes, the dynamics of the DGN model are typically rendered in a discrete iterative format:

$$D_i^t = e^{-\frac{\Delta t}{\tau_s}} D_i^{t-1} + z_i^t \tag{5}$$

$$\rho^t = \varphi(1 - g_l \cdot \Delta t - \Delta t \sum_i^N C_i D_i^t) \tag{6}$$

$$V^t = \rho^t \cdot V^{t-1} + \Delta t \sum_i^N W_i D_i^t - \vartheta z^{t-1} \tag{7}$$

$$z^t = \Theta(V^t - \vartheta) \tag{8}$$

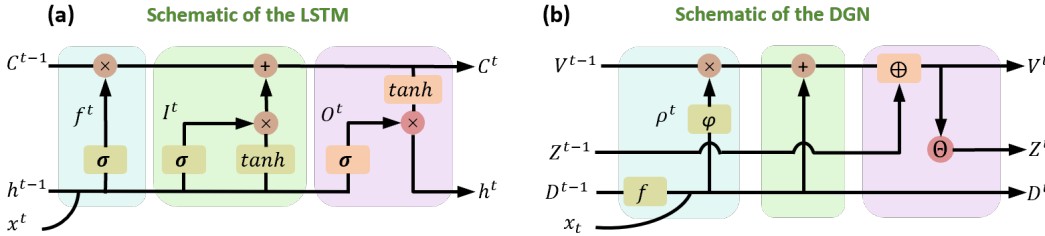

Figure 2: Schematic diagram of the model structure of LSTM and DGN. $f$: decay function. $\oplus$: reset processing.

where $\Delta t$ represents the time interval between time steps in discrete form. $\varphi$ represents numerical truncation function, such as the Sigmoid function. $\Theta$ represents Heaviside step function. An output spike $z^t$ will be generated once the membrane potential $V^t$ reaches the neuronal firing threshold $\vartheta$ as per Eq. 8. The membrane potential at the next time step will be soft reset as Eq. 7.

This study compares the biologically inspired DGN with the LIF model to elucidate their structural distinctions (Fig. 1(a-b)). Unlike the LIF model's fixed leakage conductance ($g_l$) and linear synaptic superposition, the DGN introduces dynamic conductance factors $C_i$ to establish a dual-pathway regulatory architecture. It preserves the current injection pathway ($W_i D_i$) while adding a dynamic conductance term ($C_i D_i$), forming a gated mechanism governed by $g_l + \sum C_i D_i$. This configuration can adaptively regulate the membrane potential decay rate in real time, thereby overcoming the limitations of LIF in simulating synaptic plasticity and increasing the efficiency of information transfer. This gated mechanism highlights DGN's advantages in balancing biophysical accuracy and computational performance. Appendix 4.3 demonstrates that the gating mechanism sustains excellent performance even with fewer parameters.

## 3.2 GATING STRUCTURE ANALYSIS IN CONDUCTANCE DYNAMIC SYSTEMS

The proposed Dynamic Gated Neural (DGN) model demonstrates fundamental topological homology with Long Short-Term Memory (LSTM) networks in both structural architecture and core information processing mechanisms, as illustrated in Fig. 2. Specifically, the self-adapting decay coefficient mathematically emulates the memory filtration function of LSTM's forget gate ($f^t$) (Hochreiter & Schmidhuber, 1997). The mechanism of accumulating currents through dynamic presynaptic integration is computationally similar to the input gating operation ($I^t$) (Gerstner et al., 2014). DGN's spike reset mechanism exhibits mathematical congruence with LSTM's cell state update equations, both employing nonlinear gating variables to control state transitions. This homology underscores a profound insight: the gating mechanism is a universal computational principle for regulating information flow across both artificial and biological neural systems. The LSTM model was designed with a gating mechanism to address the vanishing gradient problem in vanilla RNNs ( (Bengio et al., 1994)), while the gating mechanism of our DGN, inspired by dynamic conductance behavior in biological systems ( (Kaczmarek, 1987; Chad & Eckert, 1986)), thereby provides a more biologically grounded instantiation of this computational paradigm. DGN not only validates crucial role of gating in spatiotemporal information processing but also serves as a crucial step toward closing the long-standing gap between brain-inspired computation and artificial neural networks, providing a biologically grounded perspective on the functional origins of gating itself.

## 3.3 STABILITY OF CONDUCTANCE DYNAMIC SYSTEMS

This section establishes the theoretical framework for analyzing noise robustness in DGN model through stochastic differential equation (SDE) approaches (Mao, 2007). By linearizing the nonlinear conductance dynamics under small perturbation assumptions, we derive closed-form expressions for steady-state voltage variances in both DGN and classical LIF model. Comparative analysis of these variance solutions reveals the superior noise suppression capability of the DGN architecture.

In order to compare fairly with other models, we directly analyze the case of adding perturbations to the presynaptic input current ($D_i$ in Eq. 3). The investigation begins with stochastic input pertur-

bations modeled as Gaussian white noise superposed on deterministic signals:

$$\hat{I}_i(t) = \mu_i + \sigma_i \xi(t), \quad \langle \xi(t)\xi(t') \rangle = \delta(t - t') \tag{9}$$

where $\mu_i$ denotes deterministic input components and $\sigma_i$ quantifies noise intensity. The perturbed dynamic conductance $\hat{G}(t) = G_0 + \sum C_i \sigma_i \xi(t)$ induces voltage dynamics, where $G_0 = g_l + \sum C_i \mu_i$. The membrane potential control formula is:

$$\frac{dV}{dt} = -\hat{G}(t)V + \sum W_i \hat{I}_i(t) = \underbrace{-G_0 V + \sum W_i \mu_i}_{\text{Deterministic term}} + \underbrace{\left(-\sum C_i \sigma_i \xi(t)V + \sum W_i \sigma_i \xi(t)\right)}_{\text{Perturbation term}} \tag{10}$$

Linear noise approximation (Van Kampen, 1992) is applied by decomposing $V = V_{\text{steady}} + \delta V(t)$ with $|\delta V| \ll V_{\text{steady}}$, where $V_{\text{steady}}$ is the steady-state solution of the deterministic term. Performing Taylor expansion on nonlinear terms $C_i \sigma_i \xi(t)V$ and retaining only first-order contributions while discarding higher-order small terms ($\delta V \cdot \xi(t)$). The nonlinear perturbation term is linearized as:

$$C_i \sigma_i \xi(t)V \approx C_i \sigma_i \xi(t)V_{\text{steady}} \tag{11}$$

After truncating higher-order terms, the original SDE reduces to a linear SDE:

$$\frac{dV}{dt} = -G_0 V + \sum W_i \mu_i + \sum \sigma_i \left(W_i - C_i V_{\text{steady}}\right) \xi(t) \tag{12}$$

Using Itô calculus (Itô, 1944), the steady-state variance for DGN resolves to:

$$\langle V^2 \rangle_{\text{DGN}} = \frac{\left[\sum_{i=1}^{N} \sigma_i \left(W_i - \frac{C_i \sum_{j=1}^{N} W_j \mu_j}{G_0}\right)\right]^2}{2G_0} \tag{13}$$

For classical LIF neurons with constant leak $g_l$, the corresponding variance reduces to:

$$\langle V^2 \rangle_{\text{LIF}} = \frac{(\sum_{i=1}^{N} W_i \sigma_i)^2}{2g_l} \tag{14}$$

The derivation process of the above formula is detailed in A.1.3. Critical examination of Eq. 13 versus Eq. 14 demonstrates two synergistic noise suppression mechanisms in DGN. The denominator $G_0$ implements input-dependent leakage scaling, where intensified inputs $\mu_i$ amplify effective conductance to suppress voltage fluctuations. The numerator contains a compensatory term $\frac{C_i \sum W_j \mu_j}{G_0}$ that introduces negative feedback proportional to synaptic weights $W_i$ and coupling coefficients $C_i$. When $W_i$ and $C_i$ are positively correlated, this feedback cancels synaptic noise propagation through $W_i$, achieving partial noise rejection. In contrast, the LIF model's fixed leakage $g_l$ and absence of compensatory terms result in static noise scaling that cannot adapt to input statistics.

These analytical results quantitatively demonstrate that DGN neurons outperform LIF models in noise resilience through dynamic conductance modulation. The dual mechanism—adaptive leakage scaling and synaptic noise compensation—enables effective voltage stabilization during concurrent signal and noise processing. This theoretical framework provides fundamental insights into how conductance dynamics enhance neural computation robustness under stochastic perturbations.

## 4  EXPERIMENTS

### 4.1  COMPARISON WITH THE STATE-OF-THE-ART

Speech recognition tasks involve time-correlated contexts, making SNNs ideal due to their self-recurrent connections. To evaluate the efficiency of our DGN model, we conduct assessments on two categories of speech-related datasets: conventional audio classification benchmarks (Ti46Alpha (Mark Liberman et al., 1993) and TIDIGITS (R. Gary Leonard, 1993)) and neuromorphic speech datasets (SHD and SSC) (Cramer et al., 2020), generated through event-based encoding via CochleaAMS1b sensor processing. Details of the network architecture and training protocols are provided in Appendix A.2.2. Our experiments focus on both feedforward and recurrent SNNs with DGN model across all four datasets.

Table 1: Comparison of model performance on Ti46Alpha, TIDIGITS, SHD, and SSC datasets. Rec=N/Y represents feedforward networks (N) and recurrent networks (Y), respectively. * indicates results we reproduced using public code, while bold entries indicate the best performance.

| Datasets | Method | Rec | Hidden Layers | Accuracy(%) |
|---|---|---|---|---|
| Ti46Alpha | LIF + HM2-BP$_{\text{NeurIPS (Jin et al., 2018)}}$ | N | 800-800 | 90.98 |
| | **DGN(Ours)** | N | **100** | **95.69** |
| | RNN* | Y | 100 | 91.89 |
| | LSTM* | Y | 100 | 96.05 |
| | LIF + SrSc-SNNs-BIP$_{\text{Neural Comput. (Zhang \& Li, 2021)}}$ | Y | 400-400-400 | 95.90 |
| | **LIF + SrSc-SNNs-BIP$_{\text{Front. Neurosci. (Zhang et al., 2024b)}}$** | Y | **800** | **96.44** |
| | DGN(Ours) | Y | 100 | 96.31 |
| TIDIGITS | LIF + BAE-MPDAL$_{\text{Front. Neurosci. (Pan et al., 2020)}}$ | N | 620-11 | 97.40 |
| | LIF + Multilayer FE-Learn$_{\text{TNNLS (Luo et al., 2022)}}$ | N | 100-100 | 98.10 |
| | LIF + BPTE$_{\text{IJCNN (Lin et al., 2023)}}$ | N | 400-11 | 98.10 |
| | **DGN(Ours)** | N | **100** | **98.59** |
| | RNN* | Y | 100 | 97.09 |
| | LSTM* | Y | 100 | 97.88 |
| | **DGN(Ours)** | Y | **100** | **99.10** |
| SHD | LIF + data aug$_{\text{TNNLS (Cramer et al., 2020)}}$ | N | 128 | 49.70 |
| | TC-LIF$_{\text{AAAI, 2024 (Zhang et al., 2024a)}}$ | N | 128-128 | 83.08 |
| | **DGN(Ours)** | N | **128** | **85.18** |
| | RNN* | Y | 100 | 76.53 |
| | **LSTM$_{\text{TNNLS (Cramer et al., 2020)}}$** | **Y** | **128** | **89.20** |
| | LIF + data aug$_{\text{TNNLS (Cramer et al., 2020)}}$ | Y | 1024 | 84.50 |
| | Heterogeneous LIF$_{\text{Nat. Commun. (Perez-Nieves et al., 2021)}}$ | Y | 128 | 83.50 |
| | ALIF$_{\text{TNNLS (Yin et al., 2020)}}$ | Y | 128-128 | 84.40 |
| | TC-LIF$_{\text{AAAI (Zhang et al., 2024a)}}$ | Y | 128-128 | 88.91 |
| | DGN(Ours) | Y | 128 | 87.78 |
| | DGN(Ours) | Y | 128-128 | 88.98 |
| SSC | LIF$_{\text{TNNLS (Cramer et al., 2020)}}$ | N | 128-128 | 38.50 |
| | TC-LIF$_{\text{AAAI (Zhang et al., 2024a)}}$ | N | 128-128 | 63.46 |
| | **DGN(Ours)** | N | **128-128** | **67.54** |
| | RNN* | Y | 128-128 | 72.91 |
| | LSTM$_{\text{TNNLS (Cramer et al., 2020)}}$ | Y | 128-128 | 73.10 |
| | LIF$_{\text{TNNLS (Cramer et al., 2020)}}$ | Y | 128-128 | 52.00 |
| | Heterogeneous LIF$_{\text{Nat. Commun. (Perez-Nieves et al., 2021)}}$ | Y | 128 | 60.80 |
| | ALIF + GaussinGradient $_{\text{Nat. Mach. Intell. (Yin et al., 2021)}}$ | Y | 128 | 74.20 |
| | TC-LIF$_{\text{AAAI (Zhang et al., 2024a)}}$ | Y | 128 | 61.09 |
| | **DGN(Ours)** | Y | **128** | **66.18** |
| | **DGN(Ours)** | Y | **128-128** | **75.63** |

As shown in Tab. 1, our proposed DGN model exhibits superior performance across temporal sequential data processing tasks through its biologically inspired gating mechanism. The feedforward DGN network with a single 100-node hidden layer attains 98.59% classification accuracy on the TIDIGITS dataset, surpassing comparably structured multilayer spiking neuron networks. Notably, the dual-layer recurrent DGN achieves 75.63% accuracy on the SSC dataset, outperforming LSTM despite using fewer parameters, while both DGN and LSTM surpass other neurons lacking gating mechanisms. Our proposed DGN show excellent performance in both feedforward and recurrent networks, and their accuracy is comparable to or even better than several current SOTA methods in the field of SNNs, despite using fewer neurons and a simpler network structure. These results show that the introduced gating mechanism effectively improves the expressive power of a single neuron. This elegant solution empowers individual neurons with gated control functionality, while avoiding the incorporation of overly complex architectures. The effectiveness of the dynamic gating mechanism in enhancing the efficiency of neuronal information transmission to process complex time series data has been effectively demonstrated.

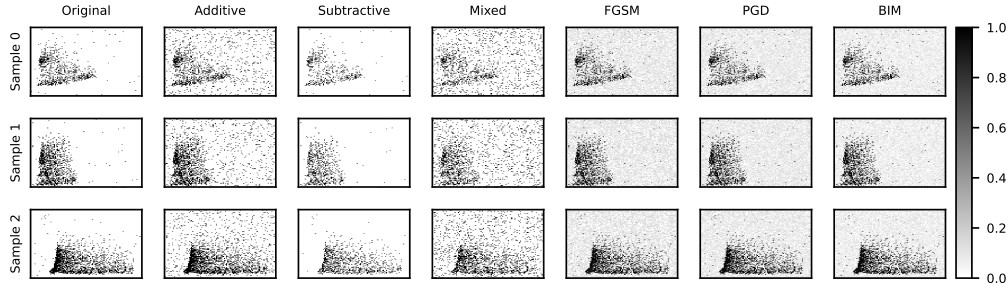

Figure 3: Noise sample visualization on SHD dataset. Each row corresponds to one SHD sample, and each column represents a noise type: Original, Additive, Subtractive, Mixed, FGSM, PGD, and BIM. The horizontal axis indicates time, and the vertical axis represents input channel indices.

Table 2: Accuracy (%) of the proposed DGN under different noise conditions and adversarial attacks on TIDIGITS and SHD. Bold entries indicate the best performance. *HeterLIF* denotes the heterogeneous LIF model proposed by Perez-Nieves et al. (2021).

| | Model | Clean | Noise | | | Attacks | | |
| | | | Additive | Subtractive | Mixed | FGSM | PGD | BIM |
|---|---|---|---|---|---|---|---|---|
| | **TIDIGITS** | | | | | | | |
| FF | LIF | 97.02 | 46.83 | 93.70 | 44.20 | 39.53 | 15.39 | 15.95 |
| | HeterLIF (Perez-Nieves et al., 2021) | 96.52 | 77.49 | 89.37 | 72.78 | 52.48 | 43.94 | 43.68 |
| | ALIF (Bellec et al., 2018) | 96.99 | 63.29 | 93.17 | 60.58 | 42.50 | 19.80 | 19.42 |
| | **DGN(Ours)** | **98.59** | **95.34** | **93.70** | **78.12** | **90.35** | **86.76** | **86.88** |
| Rec | RNN | 97.09 | 23.64 | 86.76 | 21.66 | 9.89 | 0.00 | 0.00 |
| | LSTM | 97.88 | 65.12 | 79.25 | 64.77 | 64.97 | 60.66 | 61.01 |
| | LIF | 97.80 | 73.23 | 89.60 | 67.68 | 26.55 | 61.79 | 60.70 |
| | HeterLIF (Perez-Nieves et al., 2021) | 96.29 | 78.97 | 82.59 | 73.05 | 8.76 | 36.62 | 35.74 |
| | ALIF (Bellec et al., 2018) | 97.54 | 84.01 | 86.19 | 79.25 | 25.04 | 62.82 | 63.18 |
| | **DGN(Ours)** | **99.10** | **94.84** | **96.70** | **93.86** | **89.40** | **87.52** | **87.68** |
| | **SHD** | | | | | | | |
| FF | LIF | 77.30 | 29.93 | 56.32 | 31.44 | 51.55 | 47.87 | 47.92 |
| | HeterLIF (Perez-Nieves et al., 2021) | 77.77 | 25.49 | 54.91 | 25.58 | 52.23 | 50.78 | 50.89 |
| | ALIF (Bellec et al., 2018) | 78.02 | 40.25 | 55.08 | 39.50 | 53.31 | 51.51 | 51.57 |
| | **DGN(Ours)** | **85.18** | **59.46** | **64.05** | **58.87** | **63.81** | **61.59** | **61.44** |
| Rec | RNN | 78.24 | 27.47 | 52.29 | 28.06 | 17.35 | 11.93 | 13.94 |
| | LSTM | 86.89 | 41.61 | **64.58** | 39.23 | 39.27 | 32.01 | 33.37 |
| | LIF | 75.77 | 9.24 | 57.44 | 9.25 | 17.78 | 30.59 | 31.45 |
| | HeterLIF (Perez-Nieves et al., 2021) | 79.85 | 39.57 | 58.19 | 38.87 | 44.76 | 49.12 | 49.10 |
| | ALIF (Bellec et al., 2018) | 82.08 | 46.59 | 63.32 | 47.28 | 52.2 | 58.01 | 58.31 |
| | **DGN(Ours)** | **87.78** | **78.97** | 61.91 | **79.35** | **69.45** | **66.13** | **66.34** |

## 4.2 OVERALL PERFORMANCE FOR VARIOUS PERTURBATION

To evaluate the robustness of the proposed DGN model, we implement a rigorous framework where all models are trained on pristine datasets without artificial corruption or noise. Traditional robustness evaluations typically use hybrid datasets containing noisy samples in both training and testing sets, which can lead to noise pattern memorization (Bishop, 1995; Simard et al., 2003). Traditional method introduces significant dataset construction overheads and may yield misleading assessments of noise immunity due to varying pattern learning capacities among models (Tsipras et al., 2018; Thams et al., 2022; Zhang et al., 2021). In contrast, we adopt a more challenging approach that evaluates models using previously unseen noise patterns, thereby providing a more realistic assessment of their performance under suboptimal conditions.

We considered three types of noise commonly encountered in SNNs: additive noise, subtractive noise, and mixed noise. We also evaluated model robustness under three gradient-based adversarial attacks: FGSM (Goodfellow et al., 2014), PGD (Madry et al., 2018), and BIM (Kurakin et al., 2018). We conducted anti-noise experiments on the TIDIGITS dataset and the SHD dataset to compare other models with our DGN model. Examples of how different noise types affect the input signals are shown in Fig. 3. The experimental setup is in Appendix A.2.4 and the extended experimental data is in Appendix A.5. In Tab. 2, we select sampling points of different strengths for different perturbation. Noise generation probability $p = 0.006$ for additive noise, and $p = 0.3$ for subtractive noise. The perturbation $\epsilon = 0.003$ for all attacks, and iterative step $k = 4$, step size $\alpha = 0.01$ for PGD, BIM. All results were reproduced using publicly available code to ensure a fair comparison under identical conditions. Accuracies under the clean condition can be found in Appendix A.2.3.

As shown in Tab. 2, the DGN-based feedforward network keeps 95.34% accuracy under additive noise on the TIDIGITS dataset, surpassing the conventional LIF model by 48.51%, demonstrating that its adaptive dynamic conductance learning mechanism effectively isolates noise from salient features. The robustness of gated architectures is further evidenced by comparative analyses: under PGD attacks on SHD, LSTM outperforms vanilla RNN by 20.08%, while the recurrent DGN surpasses recurrent LIF by 35.54%, validating the robustness from gated mechanisms.Across all noise conditions and adversarial attacks, the DGN model exhibits superior resistance compared to classical SNN neurons and ANNs (RNN and LSTM), maintaining the highest baseline accuracy and minimal performance degradation. These results underscore that the biologically inspired gating structure, driven by dynamic conductance modulation, fundamentally enhances robustness.

## 4.3 ABLATION STUDY

**Efficiency under Parameter Reduction.** Just as LSTM/GRU enhance MLPs by introducing complex, task-dependent structures, our DGN neuron endows SNNs with more expressive computational capabilities than traditional LIF units through the incorporation of dynamic conductance mechanisms. The observed performance improvements stem not from an increase in the number of parameters, but rather from the inherent efficiency of the neuron design. To clarify this distinction, we introduce a simplified variant, S-DGN, which reduces the parameter count while preserving the core mechanisms.

In S-DGN, all synapses connecting to the same neuron share a *single* equilibrium potential $E$, in contrast to the original DGN where each synapse maintains its own. This modification substantially reduces the number of learnable parameters, thereby constraining expressivity while preserving the essential computational mechanism. The resulting dynamics of S-DGN are formulated as follows:

$$\frac{dV}{dt} = -V \left( g_l + \sum_{i=1}^{N} C_i D_i \right) + E \sum_{i=1}^{N} C_i D_i \tag{15}$$

We evaluate its effectiveness on the SHD dataset. The evaluation considers three critical factors: classification accuracy, parameter count, and overall robustness against noise. As demonstrated in Tab. 3, s-DGN achieves competitive accuracy and noise robustness while maintaining LIF-level parameter efficiency. We demonstrate that stable performance improvements persist without increasing parameter counts, confirming that the observed benefits do not stem solely from parameter scaling but rather from the intrinsic advantages of the proposed neuronal mechanism.

**Performance under Different Perturbation Strength.** We systematically assess the robustness of spiking neuron models by measuring their classification accuracy under escalating perturbation intensities ($p$ or $\varepsilon$). As shown in Fig. 4, the proposed gated neuron model maintains higher classification accuracy with only marginal degradation when subjected to intensified noise disturbances and diverse adversarial attacks. This performance advantage is attributed to the gating mechanism's ability to dynamically adjust the neuron information transmission mode, thereby improving the model's adaptability to perturbations. Extended experiments based on other datasets (Appendix A.3.2) further and consistently demonstrate the robustness of DGN.

Table 3: Performance comparison of different neuronal models on the SHD dataset. We report the parameter count (in K), clean accuracy (%), and overall robustness (%). Highlighted rows correspond to our proposed models, **s-DGN** and **DGN**, which consistently achieve superior accuracy and robustness while demonstrating the impact of parameter reduction.

| FF Models | Parameters (K) | Clean Accuracy (%) | Overall Robustness (%) |
|---|---|---|---|
| LIF | 92.16 | 77.30 | $44.17 \pm 9.96$ |
| HeterLIF | 92.31 | 77.77 | $43.31 \pm 12.64$ |
| ALIF | 92.16 | 78.02 | $48.53 \pm 6.24$ |
| DLIF (250 step) (Ding et al., 2024a) | 92.66 | 79.96 | $49.93 \pm 5.91$ |
| s-DGN (ours) | 92.31 | 84.30 | $58.28 \pm 2.33$ |
| **DGN (ours)** | 184.32 | **85.18** | **$61.54 \pm 1.95$** |

| Rec Models | Parameters (K) | Clean Accuracy (%) | Overall Robustness (%) |
|---|---|---|---|
| LIF | 108.54 | 75.77 | $25.96 \pm 16.67$ |
| HeterLIF | 108.69 | 79.85 | $46.60 \pm 6.58$ |
| ALIF | 108.54 | 82.08 | $54.28 \pm 6.11$ |
| DLIF (250 step) (Ding et al., 2024a) | 109.04 | 82.70 | $58.87 \pm 4.64$ |
| s-DGN (ours) | 108.82 | 85.65 | $64.48 \pm 4.82$ |
| **DGN (ours)** | 217.09 | **87.78** | **$70.36 \pm 6.60$** |

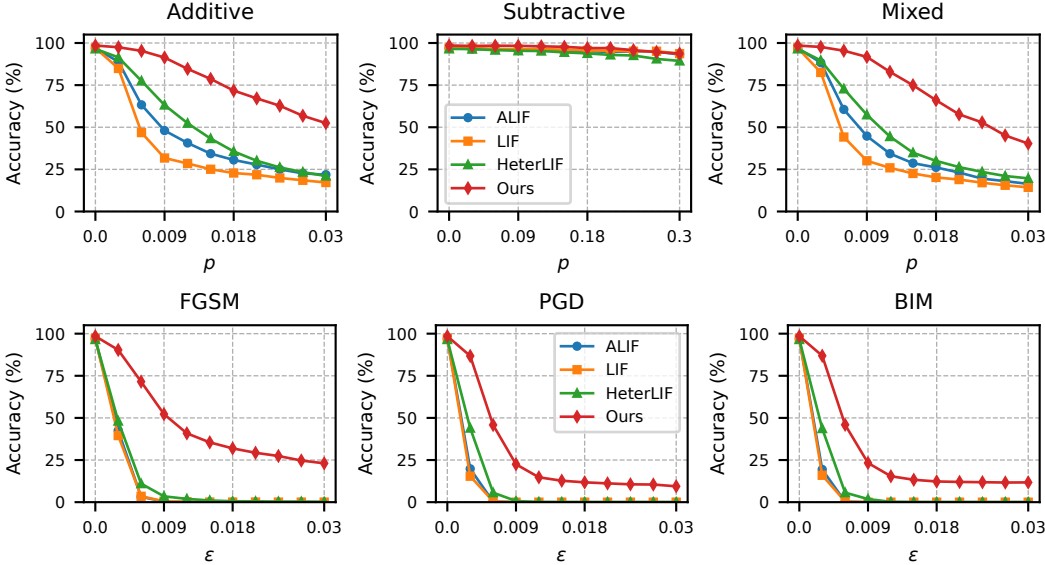

Figure 4: Performance of the model on TIDIGITS using a feedforward network under perturbations of different distribution probabilities $p$ or attack strengths $\varepsilon$.

## 5 CONCLUSION

In this work, we address the lack of bio-inspired gating mechanisms in traditional spiking neural networks (SNNs) by revisiting the biophysical principle of dynamic conductance. Inspired by the temporal behavior of biological ion channels, we propose a novel neuron model that implements a biologically inspired gating structure. This mechanism significantly enhances the spatiotemporal expressiveness and information selectivity of the neuron, leading to substantial performance improvements in speech recognition. Moreover, our model demonstrates superior robustness under various noise perturbations and adversarial attacks. This work introduces a new modeling paradigm for SNNs, offering insights into both robust computation and biologically grounded design. Future directions include integrating our approach with other advances in the SNNs community, and exploring richer conductance-based gating models with enhanced spatiotemporal properties.

## 6 ACKNOWLEDGEMENT

This work was in part supported by the Brain Science and Brain-like Intelligence Technology - National Science and Technology Major Project (Grant No. 2025ZD0215600), the National Natural Science Foundation of China (Grant No. 92370103), and in part by the Xiaomi Foundation.

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

## A APPENDIX

### A.1 METHODOLOGY

#### A.1.1 DYNAMIC GATED NEURON

For reading convenience, Eq. 2 is repeated here:

$$\frac{dg_i}{dt} = -\frac{g_i}{\tau_s} + C_i \sum_{t_i^j < t} \delta(t - t_i^j) \tag{16}$$

where $t_i^j$ indicating the arrival time of the $j$th presynaptic spike of the $i$th afferent neuron before time $t$. $\delta(x)$ represents Dirac delta function (Dirac, 1981), so $\sum_{t_i^j < t} \delta(t - t_i^j)$ is equivalent expression of $z_i(t)$. Then we solve Eq. 16 using the general solution method for first-order linear nonhomogeneous differential equations:

$$g_i(T) = e^{-\int_0^T \frac{1}{\tau_s} dt} (c + \int_0^T e^{\int_0^t \frac{1}{\tau_s} dk} C_i \sum_{t_i^j < t} \delta(t - t_i^j) dt)$$

$$= e^{-\frac{T}{\tau_s}} (c + C_i \int_0^T \sum_{t_i^j < t} e^{\frac{t}{\tau_s}} \delta(t - t_i^j) dt) \tag{17}$$

where $c \in \mathbb{R}$ is an arbitrary constant determined by the initial condition. We set $f(t, t_i^j) = e^{\frac{t}{\tau_s}} \delta(t - t_i^j)$, $\Delta t = T/n$, $d = \lceil \frac{t_i^j}{\Delta t} \rceil$ (ceiling function). Then:

$$\int_0^T \sum_{t_i^j < t} e^{\frac{t}{\tau_s}} \delta(t - t_i^j) dt = \int_0^T \sum_{t_i^j < t} f(t, t_i^j) dt$$

$$= \lim_{n \to \infty} \sum_{k=0}^n \sum_{t_i^j < k \cdot \Delta t} f(k \cdot \Delta t, t_i^j) \Delta t$$

$$= \lim_{n \to \infty} \sum_{t_i^j < T} \sum_{k=d}^n f(k \cdot \Delta t, t_i^j) \Delta t \tag{18}$$

$$= \sum_{t_i^j < T} \lim_{n \to \infty} \sum_{k=d}^n f(k \cdot \Delta t, t_i^j) \Delta t$$

$$= \sum_{t_i^j < T} \int_{t_i^j}^T f(t, t_i^j) dt$$

According to the properties of the Dirac delta function, we can get $\int_{t_i^j}^T f(t, t_i^j) dt = f(t_i^j, t_i^j) = e^{\frac{t_i^j}{\tau_s}}$. When $T = 0$, we set the conductance $g_i(T)$ of the $i$-th synapse to 0. Substituting this into Eq. 17 yields $c = 0$. So, we finally get:

$$g_i(T) = C_i \cdot e^{-\frac{T}{\tau_s}} \sum_{t_i^j < T} e^{\frac{t_i^j}{\tau_s}}$$

$$= C_i \sum_{t_i^j < T} e^{-\frac{T - t_i^j}{\tau_s}} \tag{19}$$

Then, we set $D_i^t = \sum_{t_i^j < t} e^{-\frac{t - t_i^j}{\tau_s}}$, yielding Eq. 3. Considering the discrete case in Eq. 5, we get:

$$g_i(t) = C_i \sum_j e^{-\frac{t - t_i^j}{\tau_s}} = Ci D_i^t \tag{20}$$

Substituting the above formula into Eq. 1, we can get:

$$\frac{dV}{dt} = -V\left(g_l + \sum_i^N C_i D_i\right) + \sum_i^N E_i C_i D_i \tag{21}$$

In neurobiological computational modeling, classical theoretical frameworks typically posit synaptic equilibrium potential $E_i$ as a binary-state parameter (Gütig & Sompolinsky, 2009). However, our network construction process transcend this limitation by permitting heterogeneous equilibrium potential parameters across individual synaptic units. So we set $E_i$ as a learnable parameter. The mathematical formalization method establishes the synaptic connection weight $C_i \cdot E_i$ as a learnable parameter $W_i$ through a multiplicative relationship, because $C_i$ and $E_i$ are trainable parameters. This parameterization methodology preserves biophysical interpretability while enabling multidimensional regulatory mechanisms for synaptic efficacy. Crucially, such an approach not only transcends the theoretical constraints of conventional bistable equilibrium potentials but also substantially augments the modeling capacity for network dynamics characteristics through the incorporation of continuous-spectrum $E_i$ values. So by slightly rearranging Eq. 21, we can get Eq. 5-Eq. 8.

### A.1.2 RECURRENT DGN

Our recurrent DGN follows a standard RSNN design in which each neuron receives an additional recurrent pathway driven by spike activity from the previous time step. Practically, this is implemented through an extra decaying synaptic current to the soma, without explicit self-connections. This approach is widely adopted in recurrent SNNs and extends naturally from our feedforward formulation.

The resulting update equations are:

$$D_i^t = e^{-\frac{\Delta t}{\tau_s}} D_i^{t-1} + z_i^t \tag{22}$$

$$D_{i,\text{rec}}^t = e^{-\frac{\Delta t}{\tau_s}} D_{i,\text{rec}}^{t-1} + z^{t-1} \tag{23}$$

$$\rho^t = \varphi\left(1 - g_l \Delta t - \Delta t \sum_{i=1}^N C_i D_i^t - \Delta t \sum_{i=1}^N C_{i,\text{rec}} D_{i,\text{rec}}^t\right) \tag{24}$$

$$V^t = \rho^t V^{t-1} + \Delta t \sum_{i=1}^N W_i D_i^t + \Delta t \sum_{i=1}^N W_{i,\text{rec}} D_{i,\text{rec}}^t - \vartheta z^{t-1} \tag{25}$$

$$z^t = \Theta(V^t - \vartheta) \tag{26}$$

The recurrent model is also trained using standard backpropagation through time (BPTT). In practice, we observe that the introduced recurrent pathway further enhances the computational capability and robustness of the neurons, which demonstrates that our dynamic gating mechanism generalizes well beyond purely feedforward architectures and consistently improves performance on challenging temporal tasks (as also reflected in Tab 1).And the overall robustness is also better, as shown in follow:

Table 4: Overall Robustness on the SHD dataset from Tab.2

| Datasets | Architecture | Overall Robustness(%) |
|---|---|---|
| TIDIGITS | FF | 88.56 ± 5.64 |
| | Rec | 91.67 ± 3.62 |
| SHD | FF | 61.54 ± 1.95 |
| | Rec | 70.36 ± 6.60 |

We note that the recurrent DGN does not fundamentally change the underlying neuron dynamics; rather, it provides the network with additional temporal context through recurrent synaptic currents. This recurrent integration might be one of the key reasons that lead to better performance.

In terms of training behavior, both feedforward and recurrent DGNs converge reliably under the same optimization settings. The recurrent architecture is not inherently easier to train, but its recurrent pathway allows DGN neurons to leverage their dynamic conductance mechanism more effectively, resulting in stronger temporal credit assignment and improved performance.

### A.1.3 DERIVATION OF SDE VARIANCE

The perturbation satisfies:

$$\mathbb{E}(\xi(t)) = 0, \xi(t)dt = d\mathbb{W}_t, \mathbb{E}[d\mathbb{W}_t] = 0 \tag{27}$$

where $\mathbb{W}_t$ is the Brownian motion used to describe random behavior.

We have a linear SDE of a DGN:

$$\frac{dV}{dt} = -G_0 V + \sum W_i \mu_i + \sum \sigma_i (W_i - C_i V_{\text{steady}}) \xi(t) \tag{28}$$

We take the expectation on both sides of dynamic equation Eq. 28. Then we get:

$$\frac{d}{dt}\mathbb{E}[V] = -G_0\mathbb{E}[V] + \sum W_i \mu_i + \sum \sigma_i (W_i - C_i V_{\text{steady}}) \mathbb{E}[\xi(t)] \tag{29}$$

$$= -G_0\mathbb{E}[V] + \sum W_i \mu_i \tag{30}$$

In steady state:

$$\mathbb{E}[V] = \frac{\sum W_i \mu_i}{G_0} \tag{31}$$

Applying Itô calculus (Itô, 1944) to $V^2$, we can get the calculation formula:

$$d(V^2) = 2V dV + (dV)^2 \tag{32}$$

Substituting Eq. 28 into the above equation, we get

$$d(V^2) = -2G_0 V^2 dt + (\sum 2W_i \mu_i)V dt + \left[\sum \sigma_i(W_i - C_i V_{\text{steady}})\right]^2 dt \tag{33}$$

$$+2\left[\sum \sigma_i(W_i - C_i V_{\text{steady}})\right] V\xi(t)dt \tag{34}$$

Taking the expectation on both sides, we get:

$$\frac{d\mathbb{E}[V^2]}{dt} = -2G_0\mathbb{E}[V^2] + (\sum 2W_i \mu_i)\mathbb{E}[V] + \left[\sum \sigma_i(W_i - C_i V_{\text{steady}})\right]^2 \tag{35}$$

$$+2\left[\sum \sigma_i(W_i - C_i V_{\text{steady}})\right] V\mathbb{E}[\xi(t)] \tag{36}$$

Substituting Eq. 31 into the above equation, we can get the following when taking steady state:

$$\langle V^2 \rangle = \mathbb{E}[V^2] - (\mathbb{E}[V])^2 = \frac{\left[\sum_{i=1}^{N} \sigma_i \left(W_i - \frac{C_i \sum_{j=1}^{N} W_j \mu_j}{G_0}\right)\right]^2}{2G_0} \tag{37}$$

Similarly, the steady-state variance of LIF neurons can be obtained.

### A.1.4 TRAINING DGN-SNNs WITH BPTT

The network outputs at each timestep $t$ are given by $o_t = W_L z_L^t$. Classification is based on the average of these outputs across all timesteps, computed as $y_{pred} = \frac{1}{T}\sum_{t=1}^{T} o_t$. The loss function $\mathcal{L}$ is defined over averaged outputs and is typically formulated as $E = \ell(y_{pred}, y)$, where $y$ represents the true labels and $\ell$ could be the cross-entropy function, as noted in various studies (Zheng et al., 2021; Meng et al., 2023; Fang et al., 2023; Wang et al., 2023)

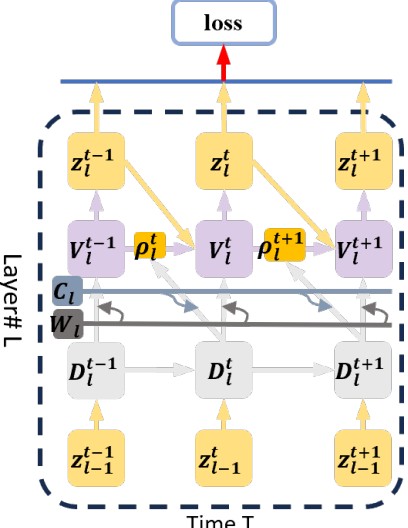

Figure 5: DGN unfolds over three time steps

BPTT unfolds the iterations described in Eq. 7, and propagates gradients back along the computational graphs across both temporal and spatial dimensions, as illustrated in Fig. 5. Subsequently, the weight update for single layer is determined among all timesteps $T$:

$$\frac{dE}{dW_i} = \sum_t^T \frac{dE}{dz^t} \frac{dz^t}{dW_i} \tag{38}$$

$$\frac{dE}{dC_i} = \sum_t^T \frac{dE}{dz^t} \frac{dz^t}{dC_i} \tag{39}$$

$$\frac{dE}{dz^t} = \frac{\partial E}{\partial z^t} + \sum_{k=t+1}^{T} [\prod_{j=t+1}^{k} (-\vartheta\Psi^j)\frac{\partial E}{\partial z^k}] \tag{40}$$

$$\frac{dz^t}{dW_i} = \Psi^t\{D_i^t + \sum_{k=1}^{t-1}[\prod_{j=k+1}^{t} (\rho^j - \vartheta\Psi^{j-1})D_i^k]\} \tag{41}$$

$$\frac{dz^t}{dC_i} = \Psi^t\{-f'V^{t-1}D_i^t + \sum_{k=1}^{t-1}[\prod_{j=k+1}^{t} (\rho^j - \vartheta\Psi^{j-1})(-f'V^{k-1}D_i^k)]\} \tag{42}$$

where:

- $\Psi^t$: surrogate gradient

- $F'$: derivative of the truncated function $\phi$ in Eq. 6

- $f'$: the value of $F'$ at $1 - (g_l + \sum_i^N C_i D_i^t)$, i.e. $f' = F'(1 - (g_l + \sum_i^N C_i D_i^t))$

In the process of gradient propagation using BPTT, it is also necessary to manually set surrogate function to calculate surrogate gradient $\Psi^t$, which are used as $dz^t/dV^t$, that is:

$$\Psi^t = \frac{dz^t}{dV^t} \tag{43}$$

The detailed derivation process of Eq. 38 ~ Eq. 42 is as follows. The gradient of the loss function $E$ with respect to the trainable weights $C_i$ and $W_i$ of synapse $i$ is:

$$\frac{dE}{dW_i} = \sum_t^T \frac{dE}{dz^t}\frac{dz^t}{dW_i} \tag{44}$$

$$\frac{dE}{dC_i} = \sum_t^T \frac{dE}{dz^t}\frac{dz^t}{dC_i} \tag{45}$$

Combining the calculation graph, we can obtain

$$\frac{dE}{dz^t} = \frac{\partial E}{\partial z^t} + \frac{dE}{dz^{t+1}}\frac{dz^{t+1}}{dz^t} \tag{46}$$

Then:

$$\frac{dz^{t+1}}{dz^t} = \frac{dz^{t+1}}{dV^{t+1}}\frac{\partial V^{t+1}}{\partial z^t} \tag{47}$$

According to Eq. 8 and Eq. 43, we obtain respectively:

$$\frac{\partial V^{t+1}}{\partial z^t} = -\vartheta \tag{48}$$

$$\frac{dz^{t+1}}{dV^{t+1}} = \Psi^{t+1} \tag{49}$$

By combining the above formula and substituting Eq. 47 into Eq. 46, we obtain:

$$\frac{dE}{dz^t} = \frac{\partial E}{\partial z^t} - \vartheta\Psi^{t+1}\frac{dE}{dz^{t+1}} \tag{50}$$

To carry out the analysis, for any time $1 \le t \le T$, we expand the recursion:

$$\frac{dE}{dz^t} = \frac{\partial E}{\partial z^t} + \sum_{k=t+1}^{T} \left[ \prod_{j=t+1}^{k} (-\vartheta\Psi^j)\frac{\partial E}{\partial z^k} \right] \tag{51}$$

Combined with the calculation graph, we get:

$$\frac{dz_t}{dW_i} = \frac{dz^t}{dV^t}\frac{dV^t}{dW_i} \tag{52}$$

$$\frac{dz_t}{dC_i} = \frac{dz^t}{dV^t}\frac{dV^t}{dC_i} \tag{53}$$

According to the calculation diagram of DGN over time, combined with Eq. 7, we can get:

$$\frac{dV^t}{dW_i} = \frac{\partial V^t}{\partial W_i} + \frac{\partial V^t}{\partial V^{t-1}}\frac{dV^{t-1}}{dW_i} + \frac{\partial V^t}{\partial z_{t-1}}\frac{dz^{t-1}}{dW_i} \tag{54}$$

$$\frac{dV_t}{dC_i} = \frac{\partial V^t}{\partial \rho^t}\frac{d\rho^t}{dC_i} + \frac{\partial V^t}{\partial V^{t-1}}\frac{dV^{t-1}}{dC_i} + \frac{\partial V^t}{\partial z^{t-1}}\frac{dz^{t-1}}{dC_i} \tag{55}$$

Substitute Eq. 52 into Eq. 54, Eq. 53 into Eq. 55, and arrange them to get:

$$\begin{aligned}\frac{dV^t}{dW_i} &= \frac{\partial V^t}{\partial W_i} + \frac{\partial V^t}{\partial V_{-1}}\frac{dV^{t-1}}{dW_i} + \frac{\partial V^t}{\partial z_{t-1}}\frac{dz^{t-1}}{dV^{t-1}}\frac{dV^{t-1}}{dW_i} \\ &= \frac{\partial V^t}{\partial W_i} + \left(\frac{\partial V^t}{\partial V_{-1}} + \frac{\partial V^t}{\partial z_{t-1}}\frac{dz^{t-1}}{dV^{t-1}}\right)\frac{dV^{t-1}}{dW_i}\end{aligned} \tag{56}$$

$$\frac{dV^t}{dC_i} = \frac{\partial V^t}{\partial \rho^t}\frac{d\rho^t}{dC_i} + \frac{\partial V^t}{\partial V^{t-1}}\frac{dV^{t-1}}{dC_i} + \frac{\partial V^t}{\partial z^{t-1}}\frac{dz^{t-1}}{dV^{t-1}}\frac{dV^{t-1}}{dC_i}$$
$$= \frac{\partial V^t}{\partial \rho^t}\frac{d\rho^t}{dC_i} + (\frac{\partial V^t}{\partial V^{t-1}} + \frac{\partial V^t}{\partial z^{t-1}}\frac{dz^{t-1}}{dV^{t-1}})\frac{dV^{t-1}}{dC_i} \qquad (57)$$

According to the Eq. 5∼8, we get:

$$\frac{\partial V^t}{\partial W_i} = D_i^t \qquad (58)$$

$$\frac{\partial V^t}{\partial V_{t-1}} = \rho^t \qquad (59)$$

$$\frac{\partial V^t}{\partial z^{t-1}} = -\vartheta \qquad (60)$$

$$\frac{dz^t}{dV^t} = \Psi^t \qquad (61)$$

$$\frac{dz^{t-1}}{dV^{t-1}} = \Psi^{t-1} \qquad (62)$$

$$\frac{\partial V^t}{\partial \rho_t} = V^{t-1} \qquad (63)$$

$$\frac{d\rho^t}{dC_i} = -F'(1-(g_l + \sum_i^N C_i D_i^t))D_i^t \qquad (64)$$

Substitute the above formula into Eq. 56, Eq. 57 and sort it out to get:

$$\frac{dV^t}{dW_i} = (\rho^t - \vartheta\Psi^{t-1})\frac{dV^{t-1}}{dW_i} + D_i^t \qquad (65)$$

$$\frac{dV^t}{dC_i} = (\rho^t - \vartheta\Psi^{t-1})\frac{dV^{t-1}}{dC_i} - f'V^{t-1}D_i^t \qquad (66)$$

Expand the recursive calculation of equations Eq. 65 and Eq. 66, and we get Eq. 38 - Eq. 42

### A.1.5 PSEUDOCODE FOR DGN COMPUTATION

We have now included pseudocode detailing the core computational steps of DGN—membrane update, dynamic conductance computation, and spiking—which aligns with Eq.5–Eq.8 and is compatible with mainstream deep-learning frameworks.

---

**Algorithm 1** Hidden Layer States Update

---

**Input:** Input spike sequence: $z_t$, Hidden layer states: $h_i, h_v, h_z$
**Output:** Updated hidden layer states: pre_i, $v, z$
  1: pre_i $\leftarrow \rho_s \cdot h_i + z_t$          ▷ Calculate presynaptic current
  2: $g \leftarrow$ pre_i $\otimes C$      ▷ Multiply by weight matrix to get dynamic conductance
  3: post_i $\leftarrow$ pre_i $\otimes W$      ▷ Multiply by weight matrix to get somatic input current
  4: $v \leftarrow \phi(1 - \mathrm{dt} \cdot g_l - \mathrm{dt} \cdot g) \cdot h_v +$ post_i $- \vartheta \cdot h_z$ ▷ Neuronal potential decay, update membrane potential
  5: $z \leftarrow \Theta(v - \vartheta)$          ▷ Calculate spike emission
  6: **return** pre_i, $v, z$

---

## A.2 EXPERIMENTS

### A.2.1 DATASETS

**TI46Alpha**: TI46Alpha is the full alphabets subset of the TI46 Speech corpus (Mark Liberman et al., 1993) and contains spoken English alphabets from 16 speakers. There are 4,142 and 6,628

spoken English examples in 26 classes for training and testing, respectively. The threshold encoding mechanism (Gütig & Sompolinsky, 2009) is used to encode the audio information into spike pattern. First, a spectrogram is computed with consecutive Fourier transforms (legacy function) from the original sound wave. Then the spectrogram is filtered by a filter bank of 16 triangular filters to obtain a mel-scale spectrogram. Next, for each mel-scale spectrogram bin corresponding to a filter, 30 neurons are used to encode its energy changes as spikes. Thus, a total of 480 neurons are used to encode an audio sample (more details, see (Gütig & Sompolinsky, 2009)). In order to increase the generalization ability of the model, we added 20 empty channels, each original audio has been converted into spike trains over 500 input channels.

**TIDIGITS**: TIDIGITS is a widely used speech recognition dataset that contains the utterances of 11 words from the digits "zero" to "nine" and "oh." It contains a training set of 2464 samples and a test set of 2486 samples. The same preprocessing used for TI46Alpha is adopted.

**SHD**: The Spiking Heidelberg Digits dataset is a spike based sequence classification benchmark, consisting of spoken digits from 0 to 9 in both English and German (20 classes). The dataset contains recordings from twelve different speakers, with two of them only appearing in the test set. Each original waveform has been converted into spike trains over 700 input channels. The train set contains 8,332 examples, and the test set consists of 2,088 examples (no validation set). In our experiments, we reduce the time resolution to speed up the simulation. Therefore, the preprocessed samples only have about 250 time steps. We determine that a channel has a spike at a certain time step of the preprocessed sample if there's at least one spike among the corresponding several time steps of the original sample.

**SSC**: The Spiking Speech Command dataset, another spike-based sequence classification benchmark, is derived from the Google Speech Commands version 2 dataset and contains 35 classes from a large number of speakers. The original waveforms have been converted to spike trains over 700 input channels. The dataset is divided into train, validation, and test splits, with 75,466, 9,981 and 20,382 examples, respectively. The same preprocessing used for SHD is adopted.

### A.2.2 TRAINING SETUP

Table 5: Network parameters for different datasets.

| Dataset | Network | $\tau_m$ | $\tau_s$ | $\vartheta$ | $(c, w)$ |
|---------|---------|----------|----------|-------------|----------|
| Ti46Alpha | feedforward | 10.00 | 2.0 | 1.00 | $(0.01 \pm 0.005, 0.01 \pm 0.005)$ |
| | recurrent | 15.00 | 1.50 | 1.00 | $(0.01 \pm 0.005, 0.01 \pm 0.005)$ |
| TIDIGITS | feedforward | 100.00 | 1.0 | 1.00 | $(0.01 \pm 0.005, 0.001 \pm 0.0005)$ |
| | recurrent | 10.00 | 2.50 | 1.00 | $(0.01 \pm 0.005, 0.01 \pm 0.005)$ |
| SHD | feedforward | 1.00 | 0.02 | 1.00 | $(0.01 \pm 0.005, 0.01 \pm 0.005)$ |
| | recurrent | 1.00 | 0.02 | 1.00 | $(0.001 \pm 0.0005, 0.001 \pm 0.0005)$ |
| SSC | feedforward | 1.00 | 0.02 | 1.00 | $(0.01 \pm 0.005, 0.01 \pm 0.005)$ |
| | recurrent | 1.00 | 0.02 | 1.00 | $(0.01 \pm 0.005, 0.01 \pm 0.005)$ |

We train the Ti46Alpha and TIDIGITS datasets for 64 epochs utilizing the Adam optimizer. Their learning rate are set to 0.001 for both feedforward and recurrent networks. For SHD and SSC datasets, we train the models for 128 epochs using the Adam optimizer. Their learning rate are set to 0.001 as well. Unlike standard binary spike trains, the SHD dataset have been temporally preprocessed to aggregate spikes within 4ms-windows (Zhang et al., 2024a; Yin et al., 2021; Cramer et al., 2020), resulting in integer spike counts per time step. We train all of the datasets on Nvidia GeForce RTX 4060 GPUs with 8GB memory for feedforward network and Nvidia GeForce RTX 4090 GPUs with 24GB memory for recurrent network.

We summarize the specific hyperparameter settings for all neuron models (including our proposed DGN and the reproduced baselines) in Tab. 5. These settings include the membrane time constant ($\tau_m$), synaptic time constant ($\tau_s$), and spike threshold ($\vartheta$). In addition, $(c, w)$ denote the initial values of the trainable parameters $(C, W)$, where $c$ is used only in the DGN model.

To eliminate confounding factors and clearly attribute performance improvements to the dynamic conductance gating mechanism itself, we evaluated models using relatively simple feedforward and recurrent architectures, and correspondingly selected baseline SNN models with comparable structural complexity. Additionally, to avoid the confounding effects of complex components—advanced spiking Transformer architectures introduce structural elements such as multi-head attention and normalization modules, which may overshadow or obscure the independent contribution of dynamic conductance gating and reduce the interpretability of mechanistic analysis—we did not include such architectures in this study. Meanwhile, we fully acknowledge that integrating dynamic conductance with spiking Transformers constitutes a promising research direction.

### A.2.3 ACCURACIES UNDER THE CLEAN CONDITION

We reproduce the results on the datasets following the referenced paper, as shown in Tab. 6. To ensure fairness in the subsequent experiments involving noise and adversarial attacks, we adopt the same hyperparameter settings across all runs to obtain the base models. The discrepancy between our reproduced accuracies and those reported in the original paper may partly stem from differences in experimental platforms. Nevertheless, conducting all evaluations under identical conditions makes our comparisons more consistent and meaningful.

For the TI46Alpha and TIDIGITS datasets, we use a single hidden layer with 100 neurons, while for the SHD and SSC datasets, we use a single hidden layer with 128 neurons.

We conducted three runs of the DGN model using the same initialization but different random seeds. As shown in Tab. 6, the first row reports the best result among the three runs, while the second row presents the mean and standard deviation. The best-performing model was then used for the noise robustness experiments.

Table 6: Accuracy of each method we reproduced on different datasets without noise or attacks.

|  | Method | Ti46alpha | Tidigits | SHD | SSC |
|---|---|---|---|---|---|
| FF | LIF | 94.00 | 97.02 | 77.30 | 47.72 |
|  | HeterLIF | 93.50 | 96.52 | 76.76 | 55.59 |
|  | ALIF | 93.85 | 96.99 | 78.02 | 49.17 |
|  | DGN(Ours) | 95.69 (95.60 ± 0.08) | 98.91 (98.24 ± 0.34) | 85.18 (84.6 ± 0.42) | 58.77 (58.34 ± 0.06) |
| Rec | RNN | 91.89 | 97.09 | 78.24 | 72.91 |
|  | LSTM | 96.05 | 97.88 | 86.89 | 75.95 |
|  | LIF | 90.89 | 97.80 | 75.77 | 53.16 |
|  | HeterLIF | 91.31 | 96.29 | 79.85 | 57.47 |
|  | ALIF | 90.28 | 97.54 | 82.08 | 55.96 |
|  | DGN(Ours) | 96.31 (95.74 ± 0.35) | 99.10 (98.67 ± 0.26) | 87.78 (86.33 ± 0.58) | 66.18 (65.72 ± 0.27) |

### A.2.4 NOISE SETUP

**Additive**
Each element of the input tensor is independently perturbed by adding a random binary value sampled from a Bernoulli distribution with probability $p$. This operation is mathematically equivalent to injecting Poisson noise with a rate of $p/\Delta t$ (Hz) into the dataset. The resulting noise tensor has the same shape as the input tensor, and the perturbed values are truncated to ensure non-negativity.

**Subtractive**
Each non-zero element in the input tensor is independently perturbed by subtracting a random binary value sampled from a Bernoulli distribution with probability $p$. The perturbation only occurs where the original data is greater than zero. After the subtraction, the resulting values are clamped from below to ensure no negative values remain.

---

**Algorithm 2** Additive Noise

---

**Input:** Data tensor $data$; Probability $p$
**Output:** Perturbed data with Bernoulli noise
 1: $noise \leftarrow \text{Bernoulli}(p)$ samples with same shape as $data$
 2: $perturbed\_data \leftarrow data + noise$
 3: $perturbed\_data \leftarrow \max(perturbed\_data, 0)$
 4: **return** $perturbed\_data$

---

**Algorithm 3** Subtractive Noise

---

**Input:** Data tensor $data$; Probability $p$
**Output:** Perturbed data with Bernoulli noise
 1: $noise \leftarrow \text{Bernoulli}(p)$ samples with same shape as $data$
 2: $mask \leftarrow (data > 0)$
 3: $perturbed\_data \leftarrow data - noise \times mask$
 4: $perturbed\_data \leftarrow \max(perturbed\_data, 0)$
 5: **return** $perturbed\_data$

---

**Mixed**

This approach combines both additive and subtractive Bernoulli noise. For non-zero elements, noise is subtracted with a higher probability scaled by a factor (default 10×). For zero elements, noise is added with the original probability $p$. All perturbations are performed independently, and the result is clamped to ensure no negative values remain. Since the input non-zero valid data is very sparse, only when the probability of subtractive noise is high can the interference effect be equal to (or even lower than) that of additive noise. Therefore, when constructing mixed noise, the probability of subtractive noise is magnified by 10 times.

---

**Algorithm 4** Mixed Noise

---

**Input:** Data tensor $data$; Probability $p$; Deletion scale factor $\gamma$ (default: 10)
**Output:** Perturbed data with mixed Bernoulli noise
 1: $delete\_mask \leftarrow (data > 0)$
 2: $delete\_p \leftarrow \min(p \times \gamma, 1)$
 3: $delete\_noise \leftarrow \text{Bernoulli}(delete\_p)$
 4: $add\_mask \leftarrow (data == 0)$
 5: $add\_noise \leftarrow \text{Bernoulli}(p)$
 6: $perturbed\_data \leftarrow data - delete\_mask \times delete\_noise + add\_mask \times add\_noise$
 7: $perturbed\_data \leftarrow \max(perturbed\_data, 0)$
 8: **return** $perturbed\_data$

---

Given a classification model $f$ with dataset $(x, y_{\text{true}})$, where $x$ is the clean image and $y_{\text{true}}$ is the corresponding correct label. The formulations of the attacks we used in this study are described as follows:

**FGSM**

FGSM aims to perturb the original data $x$ along the sign direction of the gradient on the loss function with one step to increase the perturbed linear output, thereby misleading the network, and it can be formalized as follows:

$$\hat{x} = x + \epsilon \cdot \text{sign}(\nabla_x \mathcal{L}(f(x), y_{\text{true}})), \tag{67}$$

where $\text{sign}(\cdot)$ is an odd mathematical function that extracts the sign of a real number.

**PGD**

PGD attack is the iterative variant of FGSM. It first starts from a random perturbation in the $L_p$-norm constraint around the original sample $x$, then takes a gradient iteration step in the sign direction to achieve the greatest loss output, it can be formalized as follows:

$$\hat{x}^0 = x + \mathcal{U}(-\epsilon, +\epsilon), \tag{68}$$

$$\hat{x}^{k+1} = \text{Clip}_{x,\epsilon} \left\{ \hat{x}^k + \alpha \cdot \text{sign}(\nabla_{\hat{x}^k} \mathcal{L}(f(\hat{x}^k), y_{\text{true}})) \right\}, \tag{69}$$

where $k$ is the iterative step, $\alpha$ is step size for each attack iteration, $\epsilon$ controls the perturbation level. $\mathcal{U}(\cdot)$ is a uniform function, $\text{Clip}_{x,\epsilon}\{\cdot\}$ is the function which performs per-pixel clipping of the image $\hat{x}$, so the result will be in $L_\infty$-norm $\epsilon$-neighborhood of the original image $x$.

**BIM**

Both BIM and PGD attacks are iterative attacks. Different from PGD attacks, BIM updates the adversarial samples starting from the original image.

### A.3 ABLATION STUDY

#### A.3.1 PRACTICAL SCENARIOS FOR S-DGN VS. FULL DGN

Specifically, s-DGN's reduced parameter count facilitates deployment on memory-constrained devices, rendering it well-suited for edge computing scenarios—including IoT terminals (e.g., smart sensors, edge gateways), portable AI devices (e.g., wearable health monitors, small UAV flight control systems), and low-power embedded systems with limited computational resources and video memory. In contrast, the full DGN implements the complete dynamic conductance mechanism, fully leveraging its gating effect to exhibit superior performance in processing dynamic temporal information. It is thus better tailored to scenarios demanding high performance and abundant hardware resources, such as real-time high-resolution video analysis (e.g., intelligent security early warning, autonomous driving environmental perception), complex physiological signal decoding (e.g., high-precision brain-computer interface control, real-time multi-parameter monitoring of critically ill patients), and large-scale dynamic data prediction (e.g., financial high-frequency trading analysis, smart grid dynamic load dispatching).

The full DGN model proposes a general neuronal model to implement this dynamic conductance gating mechanism. Like s-DGN, indicators such as parameter count, energy efficiency, and inference latency can be mitigated through hyperparameter adjustment or other optimization methods (e.g., model compression, hardware-aware design).

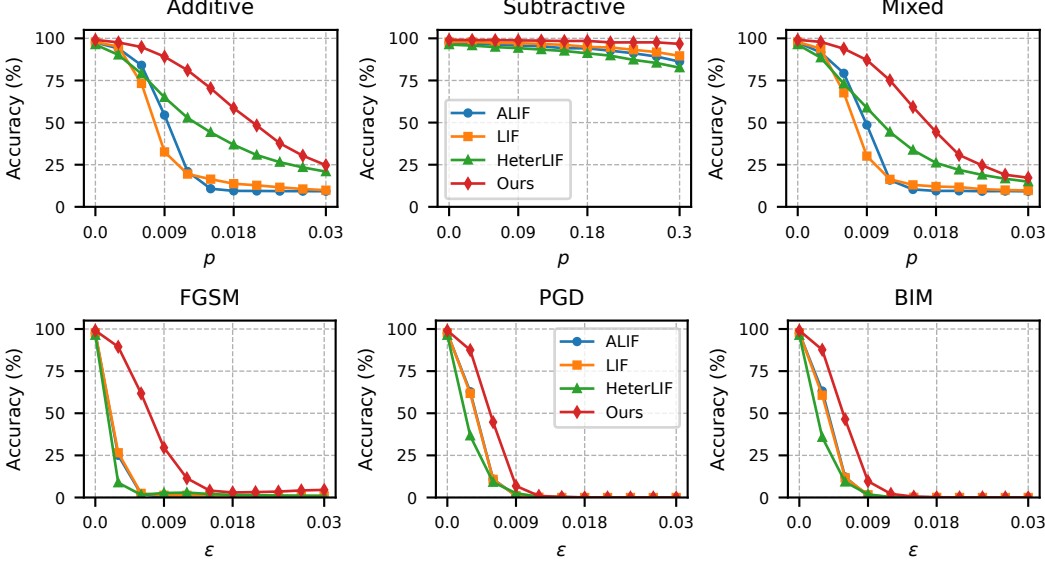

Figure 6: Performance of the model on TIDIGITS using a recurrent network under perturbations of different distribution probabilities $p$ or attack strengths $\varepsilon$.

#### A.3.2 PERFORMANCE UNDER DIFFERENT PERTURBATION STRENGTH

We systematically evaluate the robustness of multiple spiking neuron models on TIDIGITS and SHD datasets using both feedforward and recurrent networks by quantifying the performance loss under gradually increasing parameter perturbations($p$ or $\varepsilon$).

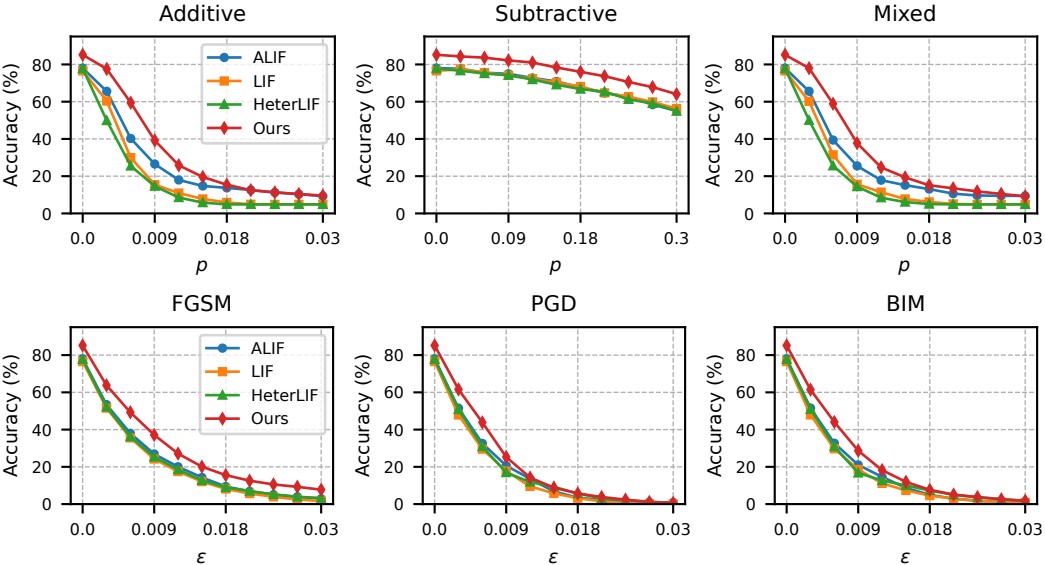

Figure 7: Performance of the model on SHD using a feedforward network under perturbations of different distribution probabilities $p$ or attack strengths $\varepsilon$.

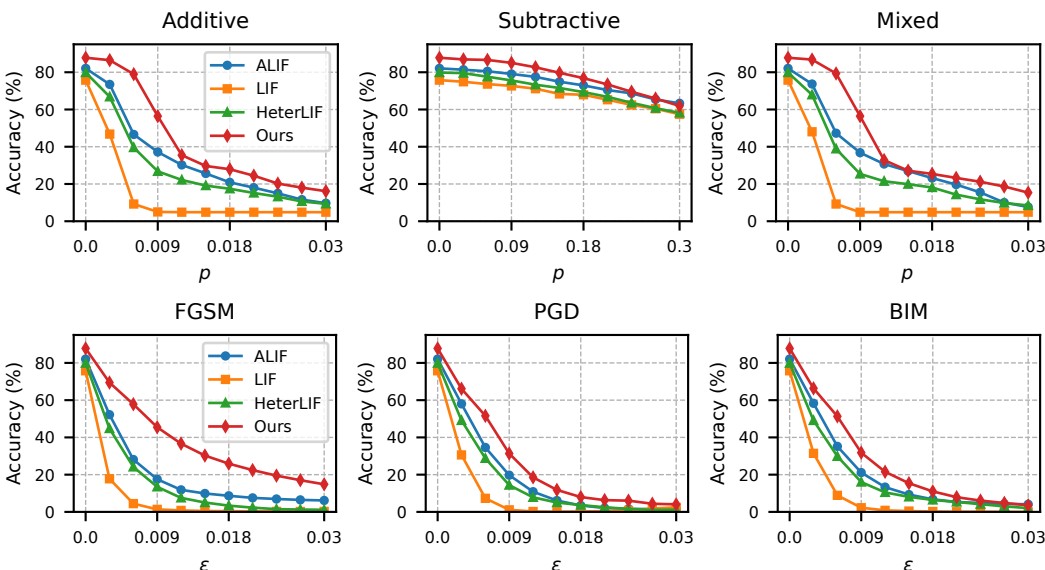

Figure 8: Performance of the model on SHD using a recurrent network under perturbations of different distribution probabilities $p$ or attack strengths $\varepsilon$.

As illustrated in Fig. 6-Fig. 8, our neuron model outperforms others in terms of accuracy under increasing perturbation intensities. It consistently maintains the highest accuracy and exhibits the lowest degradation across within a reasonable perturbation range. For example, under an additive noise perturbation of $p = 0.004$ (Poisson rate of 1 Hz), the performance loss on the SHD dataset remains within 15% for all network architectures. These results demonstrate that our neuron model is able to filter out interfering information while maintaining excellent effective information transfer efficiency, highlighting the effectiveness of the proposed bio-inspired gating mechanism in enhancing the model's robustness to a variety of perturbation patterns.

## A.4 EFFICIENCY ANALYSIS

### A.4.1 EVALUATION OF ENERGY EFFICIENCY

Energy efficiency and hardware deployment are indeed important future directions. We provide a complete analysis for transparency and practical reference as Tab. 7 and Tab. 8. For the empirical estimate, we evaluate a two-layer network (128–128 hidden units, i.e., m = n = 128) on the SHD dataset. We measure the layer-wise firing rates of the LIF and DGN neurons over 32 epochs, obtaining: LIF: $[4.43\%, 0.54\%]$, DGN: $[7.07\%, 1.19\%]$, s-DGN: $[6.38\%, 1.41\%]$.

To obtain the total energy cost, we plug these firing rates into the above formulas and adopt the 45nm CMOS process estimates from Horowitz (2014), where the energy cost of an AC and a MAC operation are $E_{\text{AC}} = 0.9\,\text{pJ}$ and $E_{\text{MAC}} = 4.6\,\text{pJ}$, respectively. This yields the empirical energy values reported in the last column of Table 8. Although DGN and s-DGN incur slightly higher energy than LIF due to the additional gating-related operations as expected, their energy consumption remains orders of magnitude lower than that of an LSTM, thus preserving the core energy-efficiency advantage of spiking neural networks.

Table 7: Comparison of neuron dynamics and per-step energy cost of LIF, LSTM, DGN and s-DGN models. The $m$ and $n$ denote the numbers of input and output neurons. $Fr_{\text{in}}$ and $Fr_{\text{out}}$ represent the firing rate of input and output neurons. $E_{\text{AC}}$ and $E_{\text{MAC}}$ are the energy cost of AC and MAC operations, respectively.

| Neuron Model | Dynamics | Step Cost |
|---|---|---|
| LIF | $V_l^t = \rho^{\text{m}} \cdot V^{t-1} + W_l^{m,n} z_l^{t-1}$ $+ W_{l,\text{rec}}^{n,n} z_{l-1}^t - \vartheta z_l^{t-1}$ | $nE_{\text{MAC}}+$ $(mnFr_{\text{in}} + nnFr_{\text{out}} + Fr_{\text{out}})nE_{\text{AC}}$ |
| DGN | $\rho_l^t = \varphi(1 - g_l - C_l^{m,n} z_{l-1}^t - C_{l,\text{rec}}^{n,n} z_l^{t-1})$ $V_l^t = \rho_l^t \cdot V_l^{t-1} + W_l^{m,n} z_{l-1}^t$ $+ W_{l,\text{rec}}^{n,n} z_l^{t-1} - \vartheta z^{t-1}$ | $mnFr_{\text{in}}E_{\text{AC}} + nnFr_{\text{out}}E_{\text{AC}}$ $nE_{\text{MAC}}+$ $(mnFr_{\text{in}} + nnFr_{\text{out}} + nFr_{\text{out}})E_{\text{AC}}$ |
| s-DGN | $\rho_l^t = \varphi(1 - g_l - C_l^{m,n} z_{l-1}^t - C_{l,\text{rec}}^{n,n} z_l^{t-1})$ $V_l^t = \rho_l^t \cdot V_l^{t-1} + (E_l C_l^{m,n}) z_{l-1}^t$ $+ (E_{l,\text{rec}} C_{l,\text{rec}}^{n,n}) z_l^{t-1} - \vartheta z^{t-1}$ | $mnFr_{\text{in}}E_{\text{AC}} + nnFr_{\text{out}}E_{\text{AC}}$ $nE_{\text{MAC}}+$ $(mnFr_{\text{in}} + nnFr_{\text{out}} + nFr_{\text{out}})E_{\text{AC}}$ |
| LSTM | $f_t = \sigma_g(W_f x_t + U_f h_{t-1} + b_f)$ $i_t = \sigma_g(W_i x_t + U_i h_{t-1} + b_i)$ $o_t = \sigma_g(W_o x_t + U_o h_{t-1} + b_o)$ $\bar{c}_t = \sigma_c(W_c x_t + U_c h_{t-1} + b_c)$ $c_t = f_t \odot c_{t-1} + i_t \odot \bar{c}_t$ $h_t = o_t \odot \sigma_h(c_t)$ | $n(m+n)E_{\text{MAC}}$ $n(m+n)E_{\text{MAC}}$ $n(m+n)E_{\text{MAC}}$ $n(m+n)E_{\text{MAC}}$ $2nE_{\text{MAC}}$ $nE_{\text{MAC}}$ |

Table 8: Total theoretical energy cost and empirical energy measurements for LIF, LSTM, DGN and s-DGN models.

| Neuron Model | Total Cost | Empirical Cost ($nJ$) |
|---|---|---|
| LIF | $(mnFr_{\text{in}} + nnFr_{\text{out}} + nFr_{\text{out}})E_{\text{AC}}$ $+ nE_{\text{MAC}}$ | 1.32 |
| DGN | $(2mnFr_{\text{in}} + 2nnFr_{\text{out}} + nFr_{\text{out}})E_{\text{AC}}$ $+ nE_{\text{MAC}}$ | 3.03 |
| s-DGN | $(2mnFr_{\text{in}} + 2nnFr_{\text{out}} + nFr_{\text{out}})E_{\text{AC}}$ $+ nE_{\text{MAC}}$ | 2.89 |
| LSTM | $(4mn + 4nn + 3n)E_{\text{MAC}}$ | 604.7 |

### A.4.2 TRAINING AND INFERENCE TIME

We also analyzed training and inference time in Tab. 9. The results, averaged over 250 training steps on the SHD dataset, show that runtime differences between models are minimal, while our approach provides superior robustness and computational capability in SNNs—our core contributions that are demonstrated in the main-text results.

Table 9: Training and Inference Time on the SHD dataset

| Arcitecture | Neuron | Train Time(s) | Test Time(s) |
|---|---|---|---|
| FF | LIF | 15.61 ± 1.49 | 5.17 ± 0.69 |
| | HeterLIF | 18.99 ± 1.80 | 5.45 ± 0.68 |
| | ALIF | 21.56 ± 2.01 | 5.80 ± 0.36 |
| | DGN(Ours) | 22.58 ± 1.48 | 5.93 ± 1.45 |
| Rec | LIF | 16.88 ± 3.39 | 5.25 ± 0.95 |
| | HeterLIF | 18.75 ± 3.17 | 5.29 ± 0.41 |
| | ALIF | 22.45 ± 4.36 | 5.58 ± 0.98 |
| | DGN(Ours) | 28.59 ± 5.00 | 6.58 ± 0.84 |

### A.4.3 DISCUSSION ON NEUROMORPHIC IMAGE DATASETS

To further evaluate the generalization ability of our proposed neuron, we extend our experiments to the neuromorphic **DVS-Gesture** dataset. This dataset features rich temporal dynamics, making it a suitable benchmark to validate whether the advantages of our design also hold beyond speech tasks.

Following prior work (Fang et al., 2021), we adopt the same network architecture to ensure fair comparison and avoid confounding factors from architectural modifications. Since our focus lies in analyzing the behavior of neurons themselves, we deliberately keep the network simple and consistent with the main text. We report preliminary reproduction results, with identical noise settings as defined in the main experiments.

Table 10: Results on the DVS-Gesture dataset. We report clean accuracy (%) and robustness under various noise and adversarial perturbations. Our proposed DGN achieves the highest accuracy and robustness across most settings.

| Neuron | Clean Acc. | Additive | Subtractive | Mixed | FGSM | PGD | BIM |
|---|---|---|---|---|---|---|---|
| LIF | 93.06 | 89.58 | 92.01 | 89.24 | 89.24 | 88.54 | 88.89 |
| HeterLIF | 94.79 | **91.67** | **92.36** | 90.97 | 86.81 | 86.46 | 85.42 |
| ALIF | 93.40 | 90.97 | 89.58 | 91.32 | 87.85 | 88.89 | 88.19 |
| DGN (ours) | **95.14** | **91.67** | **92.36** | **92.36** | **92.01** | **89.93** | **89.24** |

As shown in Table 10, our neuron consistently achieves superior accuracy and robustness, demonstrating that the proposed mechanism generalizes well beyond speech datasets and retains its advantage on image-based neuromorphic tasks.

### A.5 MORE EXPERIMENTAL RESULTS

Tab. 11-16, present the experimental results of all reproduced models (with clean accuracies shown in Tab. 6) under three types of noise and three types of adversarial attacks. In each table, the first row indicates the strength of the noise or attack. For additive and mixed noise, we vary the strength from 0 to 0.03; for subtractive noise, the range is 0 to 0.3. For all adversarial attack methods, the perturbation strength ranges from 0 to 0.003.

Table 11: Additive Noise full experiment data through different $p$.

| Net | Method | 0.003 | 0.006 | 0.009 | 0.012 | 0.015 | 0.018 | 0.021 | 0.024 | 0.027 | 0.03 |
|---|---|---|---|---|---|---|---|---|---|---|---|
| | | | | | **Datasets: Ti46alpha** | | | | | | |
| FF | LIF | 88.72 | 65.02 | 33.59 | 14.83 | 8.74 | 6.43 | 5.34 | 4.72 | 4.45 | 4.19 |
| | HeterLIF | 78.7 | 51.22 | 28.94 | 17.49 | 12.2 | 9,65 | 9 | 7.99 | 7.41 | 7.28 |
| | ALIF | 88.35 | 61.62 | 25.67 | 12.49 | 8.34 | 6.73 | 5.85 | 5.29 | 5.15 | 4.81 |
| | DGN | 92.7 | 87.21 | 77.07 | 63.77 | 52.02 | 40.65 | 31.11 | 24.62 | 19.06 | 17.02 |
| Rec | RNN | 18.8 | 7.53 | 5.2 | 4.4 | 4.24 | 4.54 | 4.56 | 4.49 | 4.84 | 4.28 |
| | LSTM | 88.65 | 64.84 | 43.16 | 30.07 | 21.63 | 16.08 | 12.46 | 10.79 | 10.44 | 9.52 |
| | LIF | 80.63 | 62.34 | 38.98 | 22.46 | 14.65 | 11.21 | 8.8 | 7.69 | 7.19 | 7.88 |
| | HeterLIF | 77.87 | 61.49 | 44.88 | 30.53 | 21.62 | 17.07 | 14.16 | 12.58 | 11.19 | 11.22 |
| | ALIF | 80.51 | 64.8 | 39.98 | 19.54 | 9.97 | 6.83 | 5.13 | 4.39 | 4.24 | 4.07 |
| | DGN | 91.86 | 81.29 | 64.58 | 44.13 | 27.68 | 17.36 | 10.3 | 7.62 | 5.81 | 4.95 |
| | | | | | **Datasets: TIDIGITS** | | | | | | |
| FF | LIF | 84.9 | 46.83 | 31.73 | 28.49 | 25.07 | 22.83 | 21.81 | 19.91 | 18.52 | 17.22 |
| | HeterLIF | 91.43 | 77.49 | 63.19 | 52.38 | 43.27 | 35.64 | 30.1 | 26.15 | 23.36 | 21.06 |
| | ALIF | 88.81 | 63.29 | 48.04 | 40.67 | 34.33 | 30.59 | 27.84 | 25 | 22.65 | 21.84 |
| | DGN | 97.47 | 95.34 | 91.36 | 86.65 | 78.67 | 71.77 | 67.12 | 62.77 | 56.8 | 52.46 |
| Rec | RNN | 35.5 | 23.64 | 19.37 | 17.56 | 16.33 | 16.43 | 16.97 | 19 | 17.74 | 15.82 |
| | LSTM | 89.49 | 65.12 | 47.8 | 29.47 | 19.83 | 16.48 | 15.16 | 13.35 | 11.95 | 12.65 |
| | LIF | 95.06 | 73.23 | 32.6 | 19.46 | 16.42 | 13.73 | 12.77 | 11.6 | 10.58 | 9.88 |
| | HeterLIF | 90.71 | 78.97 | 64.95 | 52.78 | 44.2 | 36.82 | 30.73 | 26.51 | 23.4 | 20.9 |
| | ALIF | 93.99 | 84.01 | 54.44 | 20.96 | 10.76 | 9.48 | 9.43 | 9.33 | 9.3 | 9.26 |
| | DGN | 97.56 | 94.84 | 89.12 | 81.06 | 70.38 | 58.59 | 48.28 | 37.74 | 30.34 | 24.71 |
| | | | | | **Datasets: SHD** | | | | | | |
| FF | LIF | 60.39 | 29.93 | 15.42 | 10.93 | 7.77 | 5.91 | 4.89 | 4.85 | 4.85 | 4.85 |
| | HeterLIF | 49.87 | 25.49 | 14.63 | 8.55 | 5.8 | 4.85 | 4.85 | 4.85 | 4.85 | 4.85 |
| | ALIF | 65.57 | 40.25 | 26.51 | 17.97 | 14.74 | 13.72 | 12.57 | 11.27 | 10.32 | 9.26 |
| | DGN | 77.6 | 59.46 | 39.18 | 25.86 | 19.55 | 15.43 | 12.46 | 11.29 | 10.43 | 9.56 |
| Rec | RNN | 39.26 | 27.47 | 25.27 | 24.48 | 24.15 | 22.17 | 19.92 | 16.3 | 11.38 | 7.75 |
| | LSTM | 56.09 | 41.61 | 32.2 | 29.56 | 28.82 | 27.63 | 26.7 | 25.23 | 23.36 | 21.24 |
| | LIF | 46.8 | 9.24 | 4.99 | 4.85 | 4.85 | 4.85 | 4.85 | 4.85 | 4.85 | 4.85 |
| | HeterLIF | 66.8 | 39.57 | 26.77 | 22.17 | 19.13 | 17.43 | 15.17 | 13.16 | 10.63 | 9.35 |
| | ALIF | 73.48 | 46.59 | 37.18 | 30.24 | 25.64 | 20.87 | 18.03 | 14.98 | 11.6 | 9.69 |
| | DGN | 86.49 | 78.97 | 56.41 | 35.46 | 29.67 | 27.92 | 24.44 | 20.22 | 18.04 | 16.14 |
| | | | | | **Datasets: SSC** | | | | | | |
| FF | LIF | 34.37 | 21.64 | 15.66 | 12.76 | 11 | 9.94 | 8.91 | 8.58 | 7.58 | 7.18 |
| | HeterLIF | 41.72 | 26.96 | 19.45 | 15.43 | 12.95 | 10.77 | 9.1 | 7.89 | 7.11 | 6.34 |
| | ALIF | 37.51 | 25.65 | 19.72 | 15.84 | 13.71 | 11.92 | 11.14 | 10.37 | 10.3 | 9.84 |
| | DGN | 49.62 | 31.74 | 19.47 | 13.73 | 11.15 | 9.73 | 8.67 | 7.87 | 7.26 | 6.68 |
| Rec | RNN | 57.13 | 29.15 | 19.29 | 15.68 | 14.39 | 13.74 | 13.18 | 12.34 | 11.69 | 11.03 |
| | LSTM | 60.01 | 36.53 | 23.2 | 18.26 | 16.56 | 15.65 | 14.71 | 14.23 | 13.62 | 13.15 |
| | LIF | 39.38 | 26.27 | 20.09 | 17.45 | 15.9 | 14.46 | 13.38 | 12.13 | 11.67 | 10.81 |
| | HeterLIF | 45.27 | 25.91 | 18.75 | 16.31 | 14.59 | 12.50 | 10.36 | 8.33 | 7.92 | 7.15 |
| | ALIF | 41.36 | 26.23 | 17.98 | 14.08 | 12.09 | 11.06 | 10.36 | 9.73 | 9.27 | 9.08 |
| | DGN | 45.58 | 27.62 | 18.41 | 16.55 | 15.83 | 14.96 | 14.92 | 14.34 | 13.59 | 12.19 |

Table 12: Subtractive Noise full experiment data through different $p$.

| Net | Method | 0.03 | 0.06 | 0.09 | 0.12 | 0.15 | 0.18 | 0.21 | 0.24 | 0.27 | 0.3 |
|---|---|---|---|---|---|---|---|---|---|---|---|
| | | | | | **Datasets: Ti46Alpha** | | | | | | |
| FF | LIF | 93.4 | 92.73 | 92.11 | 91.69 | 90.94 | 90.27 | 89.06 | 88.13 | 87.23 | 86.07 |
| | HeterLIF | 93 | 92.16 | 91.73 | 91.36 | 90.42 | 90.13 | 88.98 | 88.81 | 87.41 | 85.91 |
| | ALIF | 93.42 | 92.69 | 92.31 | 91.87 | 90.71 | 90.19 | 89.12 | 88.28 | 87.17 | 85.95 |
| | DGN | 95.23 | 94.81 | 94.59 | 94.08 | 93.1 | 92.81 | 91.26 | 90.45 | 88.5 | 87.18 |
| Rec | RNN | 90.94 | 90.41 | 89.13 | 88.28 | 87.01 | 85.59 | 83.32 | 80.91 | 78.01 | 74.01 |
| | LSTM | 95.5 | 95.04 | 94.29 | 93.8 | 92.51 | 90.92 | 88.14 | 84.94 | 80.6 | 75.09 |
| | LIF | 90.4 | 89.19 | 87.6 | 87.37 | 84.91 | 83.64 | 81.09 | 78.26 | 74.84 | 71.25 |
| | HeterLIF | 90.21 | 89.44 | 87.94 | 86.83 | 84.49 | 82.21 | 79.51 | 76.48 | 72.93 | 68.6 |
| | ALIF | 88.71 | 87.34 | 86.03 | 84.99 | 82.81 | 80.36 | 76.5 | 74.41 | 69.89 | 65.78 |
| | DGN | 95.74 | 95.31 | 94.48 | 94.72 | 93.79 | 93.38 | 92.43 | 91.67 | 90.72 | 89.18 |
| | | | | | **Datasets: TIDIGITS** | | | | | | |
| FF | LIF | 97.07 | 96.47 | 96.31 | 96.29 | 96.02 | 95.61 | 95.39 | 95.12 | 95.1 | 93.7 |
| | HeterLIF | 96.37 | 95.78 | 95.39 | 95.23 | 94.48 | 93.83 | 92.93 | 92.58 | 90.56 | 89.37 |
| | ALIF | 96.46 | 96.25 | 96.22 | 96.41 | 95.61 | 95.1 | 94.74 | 95 | 94.91 | 93.17 |
| | DGN | 98.27 | 98.38 | 98.31 | 98.03 | 97.72 | 96.98 | 96.98 | 95.74 | 94.63 | 93.7 |
| Rec | RNN | 96.95 | 96.72 | 96.17 | 95.66 | 95.16 | 93.87 | 92.81 | 92.12 | 90.14 | 86.76 |
| | LSTM | 97.97 | 97.77 | 97.19 | 96.42 | 95.74 | 94.57 | 91.73 | 89.07 | 84.82 | 79.25 |
| | LIF | 97.65 | 97.34 | 97.11 | 96.67 | 96.12 | 94.93 | 94.57 | 93.21 | 91.83 | 89.6 |
| | HeterLIF | 95.69 | 94.8 | 94.17 | 93.33 | 92.5 | 91.12 | 89.71 | 87.22 | 85.37 | 82.59 |
| | ALIF | 96.89 | 96.43 | 95.69 | 95.34 | 94.11 | 94.07 | 92.67 | 91.14 | 89.19 | 86.19 |
| | DGN | 98.98 | 98.93 | 98.77 | 98.63 | 98.42 | 98.48 | 97.58 | 97.6 | 97.62 | 96.7 |
| | | | | | **Datasets: SHD** | | | | | | |
| FF | LIF | 77.42 | 75.6 | 74.12 | 72.34 | 70.34 | 68.13 | 64.75 | 62.71 | 59.84 | 56.32 |
| | HeterLIF | 76.64 | 75.11 | 74.21 | 71.89 | 69.1 | 66.78 | 65.37 | 61.25 | 59.11 | 54.91 |
| | ALIF | 77.7 | 75.56 | 74.94 | 72.75 | 70.85 | 67.64 | 64.67 | 62.57 | 58.43 | 55.08 |
| | DGN | 84.3 | 83.65 | 82.17 | 81.02 | 78.35 | 76.03 | 73.69 | 70.57 | 67.83 | 64.05 |
| Rec | RNN | 75.88 | 74.41 | 72.5 | 70.83 | 67.87 | 66.71 | 62.92 | 58.96 | 55.77 | 52.29 |
| | LSTM | 86.04 | 85.34 | 84.08 | 81.98 | 80.68 | 77.49 | 75.37 | 72.45 | 69.13 | 64.58 |
| | LIF | 74.88 | 73.64 | 72.66 | 71.17 | 68.35 | 67.92 | 65.24 | 62.45 | 60.41 | 57.44 |
| | HeterLIF | 79.53 | 77.59 | 75.61 | 73.3 | 71.55 | 69.34 | 66.79 | 63.61 | 60.7 | 58.19 |
| | ALIF | 81.36 | 80.52 | 79.08 | 77.52 | 74.91 | 72.94 | 70.44 | 68.67 | 65.4 | 63.32 |
| | DGN | 86.9 | 86.61 | 85.04 | 82.67 | 79.7 | 76.84 | 73.45 | 69.53 | 65.88 | 61.91 |
| | | | | | **Datasets: SSC** | | | | | | |
| FF | LIF | 46.98 | 46.68 | 45.94 | 45.39 | 44.3 | 42.94 | 41.83 | 40.16 | 38.57 | 36.46 |
| | HeterLIF | 55.3 | 54.59 | 53.81 | 53.3 | 52.15 | 50.5 | 49.45 | 47.85 | 46.02 | 43.76 |
| | ALIF | 48.52 | 47.91 | 47.12 | 46.33 | 44.74 | 43.86 | 42.72 | 40.91 | 39.21 | 37.03 |
| | DGN | 58.51 | 58.32 | 57.87 | 57.28 | 56.21 | 54.81 | 53.37 | 51.15 | 49.27 | 47.06 |
| Rec | RNN | 72.6 | 72.25 | 71.63 | 71.21 | 70.31 | 69.24 | 68.22 | 66.75 | 65.03 | 63.29 |
| | LSTM | 75.61 | 75.47 | 74.7 | 74.12 | 73.4 | 72.25 | 71.53 | 70.06 | 68.76 | 67.5 |
| | LIF | 52.73 | 52.2 | 51.96 | 51.42 | 50.98 | 50.38 | 49.71 | 48.24 | 46.93 | 45.72 |
| | HeterLIF | 57.35 | 56.74 | 56.38 | 55.76 | 54.88 | 53.81 | 53.05 | 52.04 | 50.69 | 49.32 |
| | ALIF | 55.1 | 54.91 | 54.58 | 53.54 | 52.64 | 51.77 | 49.95 | 48.7 | 47 | 45.55 |
| | DGN | 66.32 | 66.25 | 65.47 | 64.33 | 64.14 | 62.98 | 61.71 | 60.15 | 58.99 | 57.12 |

Table 13: Mixed Noise full experiment data through different $p$.

| Net | Method | 0.003 | 0.006 | 0.009 | 0.012 | 0.015 | 0.018 | 0.021 | 0.024 | 0.026 | 0.03 |
|---|---|---|---|---|---|---|---|---|---|---|---|
| **Datasets: Ti46Alpha** | | | | | | | | | | | |
| FF | LIF | 88.17 | 60.72 | 25.61 | 10.46 | 6.78 | 5.38 | 4.9 | 4.25 | 4.12 | 4 |
| | HeterLIF | 76.49 | 45.91 | 22.95 | 13.8 | 9.58 | 8.11 | 7.39 | 6.96 | 6.11 | 6.08 |
| | ALIF | 87.62 | 56.23 | 19.87 | 9.65 | 7.22 | 5.82 | 5.28 | 4.73 | 4.57 | 4.52 |
| | DGN | 92.69 | 85.66 | 73.33 | 55.9 | 40.52 | 28.24 | 18.92 | 12.75 | 9.81 | 7.46 |
| Rec | RNN | 17.8 | 6.93 | 4.71 | 4.42 | 4.18 | 4.09 | 4.14 | 4.14 | 4.14 | 4.14 |
| | LSTM | 87.97 | 62.97 | 40.86 | 26.71 | 16.95 | 11.29 | 9.16 | 7.54 | 6.57 | 6.19 |
| | LIF | 79.03 | 57.59 | 32.31 | 18.7 | 12.19 | 8.87 | 7.32 | 6.86 | 6.51 | 6.31 |
| | HeterLIF | 75.95 | 55.96 | 36.95 | 25.09 | 16.94 | 13.34 | 11.31 | 10.64 | 9.05 | 8.06 |
| | ALIF | 79.51 | 60.85 | 34.29 | 18.02 | 8.64 | 6.38 | 4.86 | 4.39 | 4.49 | 4.08 |
| | DGN | 90.89 | 78.12 | 56.54 | 33.76 | 17.88 | 9.62 | 6.6 | 5 | 4.32 | 4.17 |
| **Datasets: TIDIGITS** | | | | | | | | | | | |
| FF | LIF | 82.54 | 44.2 | 30.1 | 25.91 | 22.58 | 20.23 | 18.96 | 17.09 | 15.56 | 14.34 |
| | HeterLIF | 89.66 | 72.78 | 57.39 | 44.48 | 34.91 | 30.04 | 26.35 | 23.49 | 21.06 | 19.67 |
| | ALIF | 88.34 | 60.58 | 44.77 | 34.33 | 28.67 | 26.16 | 23.13 | 19.52 | 18.02 | 16.37 |
| | DGN | 97.64 | 95.43 | 91.74 | 82.88 | 74.95 | 66.05 | 57.66 | 52.86 | 45.13 | 40.3 |
| Rec | RNN | 34.54 | 21.66 | 17.32 | 15.18 | 15.03 | 15.93 | 15.05 | 14.93 | 14.48 | 13.73 |
| | LSTM | 90.07 | 64.77 | 45.03 | 26.72 | 17.92 | 15.5 | 13.65 | 12.42 | 11.84 | 11.45 |
| | LIF | 93.88 | 67.68 | 30.1 | 16.31 | 13.04 | 12.04 | 11.72 | 10.48 | 9.97 | 9.73 |
| | HeterLIF | 88.5 | 73.05 | 58.73 | 44.38 | 33.68 | 26.08 | 21.93 | 18.95 | 16.69 | 14.93 |
| | ALIF | 91.64 | 79.25 | 48.57 | 15.86 | 10.33 | 9.45 | 9.49 | 9.26 | 9.35 | 9.22 |
| | DGN | 97.92 | 93.86 | 87.12 | 75.09 | 59.28 | 44.41 | 30.8 | 24.58 | 19.04 | 17.27 |
| **Datasets: SHD** | | | | | | | | | | | |
| FF | LIF | 60.18 | 31.44 | 15.67 | 11.4 | 7.77 | 6.19 | 5.09 | 4.85 | 4.85 | 4.85 |
| | HeterLIF | 50.02 | 25.58 | 14.41 | 8.44 | 6.1 | 5.02 | 4.85 | 4.85 | 4.85 | 4.85 |
| | ALIF | 65.59 | 39.4 | 25.52 | 17.84 | 15.18 | 13.08 | 10.63 | 9.63 | 9.46 | 9.41 |
| | DGN | 78.01 | 58.87 | 37.77 | 24.73 | 19.35 | 15.13 | 13.48 | 11.83 | 10.49 | 9.15 |
| Rec | RNN | 37.98 | 28.06 | 24.86 | 23.49 | 21.56 | 19.63 | 17.38 | 15.86 | 14.05 | 11.01 |
| | LSTM | 56.03 | 39.23 | 30.56 | 26.44 | 25.57 | 24.62 | 23.05 | 20.5 | 18.33 | 17.08 |
| | LIF | 48.04 | 9.25 | 4.85 | 4.85 | 4.85 | 4.85 | 4.85 | 4.85 | 4.85 | 4.85 |
| | HeterLIF | 67.89 | 38.87 | 25.4 | 21.47 | 19.89 | 18.11 | 14.28 | 11.77 | 9.8 | 8.48 |
| | ALIF | 73.68 | 47.28 | 36.81 | 30.72 | 26.93 | 23.18 | 19.69 | 15.51 | 10.02 | 7.63 |
| | DGN | 86.81 | 79.35 | 56.42 | 32.83 | 27.13 | 25.41 | 23.26 | 21.2 | 18.65 | 15.37 |
| **Datasets: SSC** | | | | | | | | | | | |
| FF | LIF | 34.67 | 21.25 | 15 | 11.69 | 10.13 | 8.97 | 8.17 | 7.33 | 6.41 | 5.8 |
| | HeterLIF | 42.05 | 26.65 | 18.36 | 14.3 | 11.66 | 9.51 | 7.88 | 6.66 | 5.73 | 5.09 |
| | ALIF | 37.74 | 25.56 | 19.03 | 14.52 | 12.6 | 10.61 | 9.53 | 8.91 | 8.35 | 7.93 |
| | DGN | 50 | 31.3 | 18.1 | 12.45 | 10.09 | 8.72 | 7.88 | 6.86 | 6.06 | 5.38 |
| Rec | RNN | 57.69 | 29.66 | 18.81 | 14.85 | 13.33 | 12.64 | 11.61 | 10.38 | 9.67 | 8.79 |
| | LSTM | 60.16 | 35.65 | 21.35 | 15.11 | 12.82 | 11.6 | 10.46 | 9.29 | 8.78 | 8.43 |
| | LIF | 39.33 | 25.98 | 19.23 | 16.5 | 14.78 | 13.32 | 12.03 | 10.66 | 9.74 | 8.73 |
| | HeterLIF | 45.13 | 25.09 | 17.54 | 14.53 | 12.31 | 9.95 | 7.31 | 5.66 | 5.23 | 5.17 |
| | ALIF | 41.87 | 26.09 | 17.71 | 13.69 | 11.59 | 10.39 | 9.55 | 8.58 | 8.26 | 7.66 |
| | DGN | 46.55 | 27.27 | 17.20 | 14.94 | 13.61 | 12.54 | 11.57 | 11.28 | 9.84 | 7.79 |

Table 14: FGSM attack full experiment data through different $\varepsilon$.

| Net | Method | 0.003 | 0.006 | 0.009 | 0.012 | 0.015 | 0.018 | 0.021 | 0.024 | 0.027 | 0.03 |
|---|---|---|---|---|---|---|---|---|---|---|---|
| | | | | | **Datasets: Ti46Alpha** | | | | | | |
| FF | LIF | 58.31 | 19.33 | 2.27 | 0.17 | 0.08 | 0.11 | 0.14 | 0.17 | 0.18 | 0.2 |
| | HeterLIF | 33.45 | 4.92 | 0.38 | 0.14 | 0.06 | 0.05 | 0.05 | 0.02 | 0.02 | 0.03 |
| | ALIF | 58.73 | 19.34 | 3.84 | 0.39 | 0.11 | 0.09 | 0.09 | 0.09 | 0.09 | 0.08 |
| | DGN | 74.74 | 50.81 | 37.85 | 30.4 | 24.96 | 21.07 | 18.29 | 16.34 | 14.66 | 13.56 |
| Rec | RNN | 0.59 | 0.41 | 0.39 | 0.38 | 0.35 | 0.33 | 0.33 | 0.33 | 0.33 | 0.33 |
| | LSTM | 50.52 | 12.16 | 6.84 | 4.74 | 3.44 | 2.93 | 2.67 | 2.44 | 2.16 | 2.1 |
| | LIF | 29.88 | 3.85 | 0.61 | 0.39 | 0.21 | 0.17 | 0.12 | 0.17 | 0.17 | 0.09 |
| | HeterLIF | 25.25 | 2.63 | 0.36 | 0.12 | 0.08 | 0.08 | 0.08 | 0.11 | 0.11 | 0.11 |
| | ALIF | 24.35 | 2.42 | 0.44 | 0.18 | 0.14 | 0.12 | 0.14 | 0.11 | 0.14 | 0.15 |
| | DGN | 52.57 | 21.17 | 10.11 | 6.41 | 5.22 | 4.61 | 4.2 | 4.15 | 3.89 | 3.52 |
| | | | | | **Datasets: TIDIGITS** | | | | | | |
| FF | LIF | 39.53 | 3.48 | 0.54 | 0.27 | 0.08 | 0.08 | 0.08 | 0.04 | 0.04 | 0.04 |
| | HeterLIF | 52.48 | 12.97 | 3.17 | 1.46 | 0.86 | 0.59 | 0.35 | 0.23 | 0.16 | 0.12 |
| | ALIF | 42.5 | 3.22 | 0.31 | 0.08 | 0.04 | 0.04 | 0.04 | 0.04 | 0.04 | 0.04 |
| | DGN | 90.35 | 71.52 | 52.25 | 40.84 | 35.49 | 31.92 | 29.31 | 27.39 | 24.71 | 23.05 |
| Rec | RNN | 9.89 | 2.59 | 0.59 | 0.16 | 0.04 | 0.04 | 0.04 | 0.04 | 0.04 | 0.04 |
| | LSTM | 64.97 | 22.48 | 9.57 | 5.3 | 4.48 | 4.25 | 3.66 | 2.32 | 2.51 | 2.32 |
| | LIF | 26.55 | 2.45 | 1.33 | 1.33 | 1.25 | 1.08 | 1.01 | 0.93 | 0.97 | 0.97 |
| | HeterLIF | 8.76 | 1.62 | 2.67 | 2.91 | 2.08 | 1.54 | 1.34 | 1.19 | 1.03 | 0.95 |
| | ALIF | 25.04 | 2.04 | 0.23 | 0.04 | 0.04 | 0.04 | 0.04 | 0.08 | 0.08 | 0.08 |
| | DGN | 89.4 | 61.91 | 29.62 | 11.48 | 4.18 | 3.08 | 3.28 | 3.55 | 4.2 | 4.51 |
| | | | | | **Datasets: SHD** | | | | | | |
| FF | LIF | 51.55 | 35.57 | 24.22 | 17.51 | 12.09 | 8.14 | 5.67 | 3.92 | 2.49 | 1.65 |
| | HeterLIF | 52.23 | 36.2 | 25.35 | 18.55 | 12.76 | 8.94 | 6.98 | 5.2 | 3.9 | 3.25 |
| | ALIF | 53.31 | 38 | 26.89 | 20.06 | 14.32 | 9.42 | 6.86 | 5.07 | 3.92 | 2.86 |
| | DGN | 63.81 | 49.29 | 37.15 | 27.12 | 20.08 | 15.55 | 12.56 | 10.55 | 9.27 | 7.73 |
| Rec | RNN | 17.35 | 6.61 | 4.03 | 3.38 | 2.84 | 2.53 | 2.1 | 1.99 | 1.91 | 1.86 |
| | LSTM | 39.27 | 19.19 | 8.38 | 3.81 | 2.04 | 1.56 | 1.52 | 1.52 | 1.34 | 1.12 |
| | LIF | 17.78 | 4.45 | 1.35 | 0.8 | 0.49 | 0.39 | 0.3 | 0.26 | 0.22 | 0.26 |
| | HeterLIF | 44.76 | 24.15 | 13.45 | 7.53 | 5.03 | 3.37 | 2.27 | 1.48 | 1.27 | 1.14 |
| | ALIF | 52.2 | 28.12 | 17.55 | 11.84 | 9.89 | 8.65 | 7.54 | 6.93 | 6.45 | 6.15 |
| | DGN | 69.45 | 57.82 | 45.46 | 36.65 | 30.25 | 25.9 | 22.45 | 19.47 | 17.03 | 14.82 |
| | | | | | **Datasets: SSC** | | | | | | |
| FF | LIF | 14.77 | 4.37 | 1.44 | 0.54 | 0.18 | 0.05 | 0.03 | 0.01 | 0 | 0 |
| | HeterLIF | 7.87 | 0.54 | 0.03 | 0.01 | 0.01 | 0.01 | 0 | 0 | 0 | 0 |
| | ALIF | 12.61 | 2.6 | 0.54 | 0.1 | 0.01 | 0 | 0 | 0 | 0 | 0 |
| | DGN | 29.44 | 12.5 | 5.58 | 2.77 | 1.58 | 0.93 | 0.59 | 0.42 | 0.33 | 0.24 |
| Rec | RNN | 20.59 | 4.96 | 2.18 | 1.31 | 0.73 | 0.46 | 0.26 | 0.19 | 0.17 | 0.14 |
| | LSTM | 29.41 | 11.2 | 7.25 | 5.9 | 4.9 | 4.19 | 3.75 | 3.24 | 2.83 | 2.55 |
| | LIF | 12.83 | 2.87 | 0.86 | 0.26 | 0.07 | 0.04 | 0.02 | 0.02 | 0.01 | 0.01 |
| | HeterLIF | 15.53 | 2.14 | 0.20 | 0.04 | 0 | 0 | 0 | 0 | 0 | 0 |
| | ALIF | 17.1 | 4.64 | 1.39 | 0.49 | 0.21 | 0.13 | 0.05 | 0.02 | 0.03 | 0.01 |
| | DGN | 30.99 | 12.63 | 6.49 | 3.99 | 2.77 | 2.05 | 1.60 | 1.26 | 0.94 | 0.69 |

Table 15: PGD attack full experiment data through different $\varepsilon$.

| Net | Method | 0.003 | 0.006 | 0.009 | 0.012 | 0.015 | 0.018 | 0.021 | 0.024 | 0.027 | 0.03 |
|---|---|---|---|---|---|---|---|---|---|---|---|
| **Datasets: Ti46Alpha** | | | | | | | | | | | |
| FF | LIF | 53.51 | 10.04 | 0.11 | 0 | 0 | 0 | 0 | 0 | 0 | 0 |
| | HeterLIF | 20.62 | 1.38 | 0.05 | 0.02 | 0 | 0 | 0 | 0 | 0 | 0 |
| | ALIF | 11.73 | 0.53 | 0.03 | 0 | 0 | 0 | 0 | 0 | 0 | 0 |
| | DGN | 68.54 | 31.97 | 20.58 | 15.52 | 11.44 | 9.48 | 8.36 | 7.82 | 6.95 | 6.64 |
| Rec | RNN | 0.02 | 0 | 0 | 0 | 0 | 0 | 0 | 0 | 0 | 0 |
| | LSTM | 41.94 | 3.09 | 0.95 | 1.05 | 1.28 | 0.91 | 0.51 | 0.26 | 0.12 | 0.03 |
| | LIF | 18.75 | 1.09 | 0.56 | 0.29 | 0.09 | 0.03 | 0.02 | 0 | 0.03 | 0.06 |
| | HeterLIF | 20.07 | 4.48 | 1.78 | 0.87 | 0.39 | 0.3 | 0.51 | 0.77 | 0.96 | 1.17 |
| | ALIF | 16.97 | 1.9 | 0.42 | 0.11 | 0.03 | 0.08 | 0.02 | 0 | 0.03 | 0.06 |
| | DGN | 39.55 | 10.06 | 3.02 | 1.21 | 0.43 | 0.54 | 0.41 | 0.3 | 0.3 | 0.3 |
| **Datasets: TIDIGITS** | | | | | | | | | | | |
| FF | LIF | 15.39 | 0.54 | 0 | 0 | 0 | 0 | 0 | 0 | 0 | 0 |
| | HeterLIF | 43.94 | 5.64 | 0.7 | 0.08 | 0 | 0 | 0 | 0 | 0 | 0 |
| | ALIF | 19.8 | 0.33 | 0 | 0 | 0 | 0 | 0 | 0 | 0 | 0 |
| | DGN | 86.76 | 45.81 | 22.49 | 14.75 | 12.72 | 11.76 | 11.15 | 10.58 | 10.47 | 9.41 |
| Rec | RNN | 0 | 0 | 0 | 0 | 0 | 0 | 0 | 0 | 0 | 0 |
| | LSTM | 60.66 | 3.71 | 0.9 | 0.64 | 0.12 | 1.12 | 0.37 | 0.16 | 0 | 0 |
| | LIF | 61.79 | 10.74 | 1.07 | 0.16 | 0 | 0 | 0 | 0 | 0 | 0 |
| | HeterLIF | 36.62 | 9.06 | 2.57 | 0.44 | 0.16 | 0.04 | 0.08 | 0 | 0.12 | 0 |
| | ALIF | 62.82 | 10.88 | 1.07 | 0.12 | 0.35 | 0.04 | 0 | 0 | 0 | 0 |
| | DGN | 87.52 | 44.74 | 6.84 | 0.98 | 0.27 | 0.08 | 0.04 | 0.04 | 0.04 | 0 |
| **Datasets: SHD** | | | | | | | | | | | |
| FF | LIF | 47.87 | 29.53 | 17.78 | 9.51 | 5.74 | 3.18 | 1.5 | 0.45 | 0.17 | 0 |
| | HeterLIF | 50.78 | 30.91 | 16.92 | 11.83 | 9.25 | 5.65 | 3.14 | 2.12 | 1.15 | 0.78 |
| | ALIF | 51.51 | 32.58 | 20.64 | 13.58 | 6.79 | 3.42 | 2.27 | 1.47 | 0.97 | 0.84 |
| | DGN | 61.59 | 43.86 | 25.3 | 14.22 | 8.81 | 5.72 | 3.79 | 2.4 | 1.14 | 0.63 |
| Rec | RNN | 11.93 | 3.52 | 2.73 | 2.08 | 2.17 | 1.38 | 0.3 | 0.22 | 0.04 | 0 |
| | LSTM | 32.01 | 9.54 | 2.64 | 1.36 | 0.61 | 0.22 | 0.26 | 0.17 | 0.04 | 0 |
| | LIF | 30.59 | 7.29 | 1.12 | 0.22 | 0.09 | 0.22 | 0.43 | 0.79 | 1.78 | 2.45 |
| | HeterLIF | 49.12 | 28.76 | 14.37 | 7.83 | 5.03 | 3.67 | 2.49 | 1.6 | 1.07 | 1.17 |
| | ALIF | 58.01 | 34.6 | 19.72 | 10.85 | 6.1 | 3.44 | 2.1 | 1.63 | 1.5 | 1.46 |
| | DGN | 66.13 | 51.59 | 31.34 | 18.56 | 11.81 | 7.97 | 6.32 | 6.04 | 4.37 | 4.05 |
| **Datasets: SSC** | | | | | | | | | | | |
| FF | LIF | 10.67 | 1.35 | 0.13 | 0 | 0 | 0 | 0 | 0 | 0 | 0 |
| | HeterLIF | 3.45 | 0.03 | 0 | 0 | 0 | 0 | 0 | 0 | 0 | 0 |
| | ALIF | 8.58 | 0.7 | 0.04 | 0 | 0 | 0 | 0 | 0 | 0 | 0 |
| | DGN | 23.64 | 5.23 | 1.6 | 0.45 | 0.05 | 0.01 | 0 | 0 | 0 | 0 |
| Rec | RNN | 16.87 | 4.19 | 1.68 | 0.38 | 0.38 | 0.07 | 0 | 0 | 0 | 0 |
| | LSTM | 29.41 | 11.2 | 7.25 | 5.9 | 4.9 | 4.19 | 3.75 | 3.24 | 2.83 | 2.55 |
| | LIF | 8.35 | 0.88 | 0.14 | 0.02 | 0 | 0 | 0 | 0 | 0 | 0 |
| | HeterLIF | 10.26 | 0.53 | 0.12 | 0.02 | 0 | 0 | 0 | 0 | 0 | 0 |
| | ALIF | 12.89 | 2.14 | 0.6 | 0.08 | 0.03 | 0 | 0 | 0 | 0 | 0 |
| | DGN | 20.55 | 5.65 | 2.22 | 0.74 | 0.21 | 0.03 | 0.02 | 0.01 | 0.01 | 0.01 |

Table 16: BIM attack full experiment data through different $\varepsilon$.

| Net | Method | 0.003 | 0.006 | 0.009 | 0.012 | 0.015 | 0.018 | 0.021 | 0.024 | 0.027 | 0.03 |
|---|---|---|---|---|---|---|---|---|---|---|---|
| | | | | | **Datasets: Ti46Alpha** | | | | | | |
| FF | LIF | 53.16 | 10.18 | 0 | 0 | 0 | 0 | 0 | 0 | 0 | 0 |
| | HeterLIF | 22.03 | 1.52 | 0.12 | 0 | 0 | 0 | 0 | 0 | 0 | 0 |
| | ALIF | 55.25 | 11.89 | 0.61 | 0.02 | 0 | 0 | 0 | 0 | 0 | 0 |
| | DGN | 68.35 | 13.98 | 0.23 | 0.02 | 0 | 0 | 0 | 0 | 0 | 0 |
| Rec | RNN | 0.02 | 0.02 | 0 | 0 | 0 | 0 | 0 | 0 | 0 | 0 |
| | LSTM | 42.92 | 3.78 | 0.86 | 0.29 | 0.18 | 0.24 | 0.08 | 0.03 | 0.08 | 0.08 |
| | LIF | 17.31 | 1.31 | 0.76 | 0.33 | 0.12 | 0.03 | 0.03 | 0.03 | 0.02 | 0.02 |
| | HeterLIF | 19.78 | 4.86 | 1.8 | 0.97 | 0.62 | 0.49 | 0.29 | 0.11 | 0.09 | 0.08 |
| | ALIF | 16.48 | 1.37 | 0.5 | 0.15 | 0.05 | 0.12 | 0.17 | 0.05 | 0.03 | 0 |
| | DGN | 39.79 | 10.54 | 3.24 | 1.33 | 0.89 | 0.51 | 0.45 | 0.32 | 0.32 | 0.33 |
| | | | | | **Datasets: TIDIGITS** | | | | | | |
| FF | LIF | 15.95 | 0.41 | 0 | 0 | 0 | 0 | 0 | 0 | 0 | 0 |
| | HeterLIF | 43.68 | 5.64 | 1.84 | 0.16 | 0.04 | 0 | 0 | 0 | 0 | 0 |
| | ALIF | 19.42 | 0.25 | 0 | 0 | 0 | 0 | 0 | 0 | 0 | 0 |
| | DGN | 86.88 | 45.93 | 23.26 | 15.39 | 13.29 | 12.27 | 12 | 11.84 | 11.65 | 11.73 |
| Rec | RNN | 0 | 0 | 0 | 0 | 0 | 0 | 0 | 0 | 0 | 0 |
| | LSTM | 61.01 | 6.06 | 1.45 | 1.29 | 0.59 | 0.66 | 0.43 | 0.27 | 0.27 | 0.27 |
| | LIF | 60.7 | 11.72 | 1.62 | 0.16 | 0.08 | 0.08 | 0.04 | 0 | 0 | 0 |
| | HeterLIF | 35.74 | 9.23 | 1.88 | 0.2 | 0.04 | 0 | 0.08 | 0.04 | 0 | 0 |
| | ALIF | 63.18 | 12.2 | 0.72 | 0.08 | 0.41 | 0.08 | 0.08 | 0.08 | 0 | 0 |
| | DGN | 87.68 | 46.45 | 9.65 | 2.15 | 0.51 | 0.16 | 0.08 | 0.08 | 0.08 | 0.08 |
| | | | | | **Datasets: SHD** | | | | | | |
| FF | LIF | 47.92 | 29.75 | 18.59 | 11.15 | 7.34 | 4.57 | 2.88 | 1.65 | 1.04 | 0.86 |
| | HeterLIF | 50.89 | 30.87 | 16.72 | 12.65 | 10.01 | 7.28 | 4.98 | 3.66 | 2.66 | 1.73 |
| | ALIF | 51.57 | 32.67 | 21.03 | 14.62 | 8.68 | 4.98 | 2.88 | 1.88 | 1.1 | 0.8 |
| | DGN | 61.44 | 44.05 | 28.66 | 18.23 | 11.93 | 7.65 | 5.09 | 3.77 | 2.54 | 1.8 |
| Rec | RNN | 13.94 | 3.93 | 2.34 | 1.27 | 0.99 | 0.86 | 0.6 | 0.45 | 0.22 | 0.17 |
| | LSTM | 33.37 | 13.49 | 5.56 | 2.12 | 0.63 | 0.09 | 0 | 0 | 0 | 0 |
| | LIF | 31.45 | 8.9 | 2.24 | 0.84 | 0.44 | 0.23 | 0.09 | 0.09 | 0.04 | 0 |
| | HeterLIF | 49.1 | 29.85 | 16.03 | 10.46 | 8.06 | 6.43 | 5.32 | 4.14 | 2.99 | 2.03 |
| | ALIF | 58.31 | 35.19 | 21.12 | 13.31 | 9.31 | 6.75 | 5.32 | 4.79 | 4.31 | 4.09 |
| | DGN | 66.34 | 51.3 | 31.8 | 21.54 | 115.4 | 11 | 7.89 | 6.07 | 4.8 | 3.55 |
| | | | | | **Datasets: SSC** | | | | | | |
| FF | LIF | 10.71 | 1.48 | 0.18 | 0.01 | 0 | 0 | 0 | 0 | 0 | 0 |
| | HeterLIF | 3.59 | 0 | 0 | 0 | 0 | 0 | 0 | 0 | 0 | 0 |
| | ALIF | 8.56 | 0.84 | 0.04 | 0 | 0 | 0 | 0 | 0 | 0 | 0 |
| | DGN | 23.72 | 5.89 | 1.9 | 0.62 | 0.13 | 0.03 | 0.01 | 0 | 0 | 0 |
| Rec | RNN | 17.23 | 4.65 | 1.81 | 0.51 | 0.15 | 0.02 | 0.01 | 0.01 | 0.01 | 0 |
| | LSTM | 18.78 | 4.08 | 1.41 | 0.46 | 0.2 | 0.13 | 0.09 | 0.06 | 0.04 | 0.04 |
| | LIF | 8.14 | 0.93 | 0.23 | 0.06 | 0.01 | 0 | 0 | 0 | 0 | 0 |
| | HeterLIF | 10.31 | 0.59 | 0.09 | 0.01 | 0 | 0 | 0 | 0 | 0 | 0 |
| | ALIF | 12.92 | 2.26 | 0.62 | 0.15 | 0.02 | 0 | 0 | 0 | 0 | 0 |
| | DGN | 21.12 | 5.93 | 2.36 | 0.96 | 0.38 | 0.17 | 0.07 | 0.06 | 0.05 | 0.03 |

