# OpenReview forum: "A Brain-Inspired Gating Mechanism Unlocks Robust Computation in Spiking Neural Networks"
_ICLR.cc/2026/Conference — ICLR 2026 Poster_

### Official Review · Reviewer_1AVk · 2025-10-30

**Soundness:** 4
**Presentation:** 4
**Contribution:** 3
**Rating:** 8
**Confidence:** 5

**Summary:**

This paper introduces the Dynamic Gated Neuron (DGN), a novel spiking neuron model that incorporates dynamic membrane conductance as a biologically-motivated gating mechanism. DGN modulates membrane decay via activity-dependent conductance, enabling selective information retention and improved robustness to noise. The authors connect DGN behavior to gating in LSTMs, analyze stochastic stability, and present empirical results on many temporal datasets, showing improved accuracy and robustness under noise and adversarial attacks.

**Strengths:**

1. The work is significant, as it draws clear inspiration from neuroscience, building on established literature on activity-dependent ion-channel plasticity, to introduce an input-dependent gating mechanism into spiking neuron models. Different from LSTM in conventional deep learning, the proposed DGN model is biologically grounded and novel. It may have a significant multi-disciplinary impact.

2. This paper is well-written and easy to follow. It presents a well-structured bridge among conductance-based neurons in computational neurosience, LSTM in deep learning, and LIF neurons in SNNs, and clearly articulates the differences and connections among DGN, LSTM, and LIF, offering valuable multi-disciplinary insight.

3. The work is supported by rigorous theoretical analysis demonstrating how dynamic conductance mechanisms enable adaptive leakage and noise suppression.

4. Experiments are extensive in this work, with strong results supporting the claims and demonstrating the strong temporal processing capability and robustness of DGN.

5. The proposed model is also advantageous in parameter efficiency, which is particularly valuable for SNNs operating in edge applications.

**Weaknesses:**

1. The authors seem not to discuss the feasibility of deploying the DGN model on neuromorphic chips. Since the computational advantages of SNNs are primarily achieved on such hardware, it would be valuable to discuss this aspect.

2. The DGN model includes a numerical truncation function. However, it is unclear which specific type of truncation function is used in the experiments. What influence does the choice of truncation function have on performance? Are these functions hardware-friendly?

**Questions:**

It is not very clear how the DGN model is used to build both feedforward and recurrent networks. From Table 1, recurrent DGNs appear to outperform their feedforward counterparts. Could the authors clarify how the recurrent version is constructed and trained? Additionally, is the recurrent DGN more stable or easier to train compared to the feedforward architecture, and if so, why?

---

> ### Author Response · Authors · 2025-11-21
>
> First, we sincerely thank the reviewer for the positive assessment of our work and for the constructive comments. Below we provide our responses to the concerns you pointed out.
>
> ### 1. About the feasibility of deploying on neuromorphic chips
>
> We fully agree with the reviewer that the computational advantages of SNNs are most evident on neuromorphic hardware, and that it is therefore important to discuss the feasibility of deploying DGN on such platforms. This is indeed an important direction of our ongoing and future work, and we have already conducted some preliminary explorations in this regard.
>
> From an implementation viewpoint, DGN differs from a standard LIF neuron (for which many hardware realizations already exist) mainly through the dynamically modulated leak factor \rho^t. In our formulation, this factor is computed as
>
> $$\rho^t = \varphi\big(1 - G^t\big), \quad  G^t = g_l \Delta t + \Delta t \sum_{i=1}^N C_i D_i^t.$$
>
> Here, the first term $g_l \Delta t$ is a fixed constant, and the second term $\Delta t \sum_{i=1}^N C_i D_i^t$ is structurally identical to a standard synaptic accumulation (a weighted sum of synaptic state variables $D_i^t$), for which current SNN neuromorphic chips (both digital and analog based platforms) already provide mature hardware support. The remaining nonlinearity $\varphi(\cdot)$ is a simple hard clipping function; we discuss its hardware friendliness in the next paragraph.
>
>
>
> ### 2. About the truncation function
>
> All experimental results reported in our paper use a simple hard clipping function, defined as:
>
> $$f(x) = \begin{cases} 1, & x > 1 \\\\ x, & 0 \le x \le 1 \\\\ 0, & x < 0 \end{cases}$$
>
> i.e.,  $f(x) = \min(\max(x, 0), 1)$
>
> In principle, different choices of bounded nonlinearities (e.g., a smooth sigmoid-like function) could slightly affect performance, and smoother functions may provide more gradual saturation near the boundaries. In our preliminary exploration, we also experimented with several alternative clipping functions (including sigmoid-based smooth saturations), but we did not observe consistent or significant performance gains. On the contrary, these alternatives introduced additional confounding factors when analyzing the effect of the dynamic gating mechanism itself. Furthermore, smoother nonlinearities tend to complicate neuromorphic deployment. For these reasons, all results reported in the paper are obtained using the simple hard clipping function defined above.
>
> From a hardware perspective, this hard clipping function maps very naturally to digital neuromorphic implementations: it can be realized with just two comparisons and multiplexers, or as standard saturating arithmetic in fixed-point units. Similar hard clipping to a fixed interval is a common practice in low-bitwidth CNN/DNN accelerators, where it is efficiently implemented using comparator–mux logic or by constraining the numerical range of the underlying fixed-point representation. This hard clipping operation corresponds exactly to standard saturation arithmetic on fixed-point hardware, which is a very mature and widely used primitive in digital signal processing and low-bitwidth neural network accelerators.
>
> Taken together, these observations indicate that the proposed DGN model is well suited for deployment on neuromorphic hardware, and we are already actively pursuing this direction in our ongoing work.

---

> ### Author Response · Authors · 2025-11-21
>
> ### 3. About recurrent networks
>
> We thank the reviewer for requesting clarification. Our recurrent DGN follows a standard RSNN design in which each neuron receives an additional recurrent pathway driven by spike activity from the previous time step. Practically, this is implemented through an extra decaying synaptic current to the soma, without explicit self-connections. This approach is widely adopted in recurrent SNNs and extends naturally from our feedforward formulation.
>
> The resulting update equations are:
>
> $$\begin{gather}    D_i^t = e^{-\frac{\Delta t}{\tau_{\text{s}}}} D_i^{t-1} + z_i^t \\\\[2mm]    D_{i,\text{rec}}^t = e^{-\frac{\Delta t}{\tau_{\text{s}}}} D_{i,\text{rec}}^{t-1} + z^{t-1} \\\\[2mm]    \rho^t = \varphi\!\Big(1 - g_l \Delta t - \Delta t \sum_{i=1}^N C_i D_i^t - \Delta t \sum_{i=1}^N C_{i,\text{rec}} D_{i,\text{rec}}^t\Big) \\\\[2mm]    V^{t} = \rho^t V^{t-1} + \Delta t \sum_{i=1}^{N} W_i D_i^t + \Delta t \sum_{i=1}^N W_{i,\text{rec}} D_{i,\text{rec}}^t - \vartheta z^{t-1} \\\\[2mm]    z^t = \Theta(V^t - \vartheta). \end{gather}$$
>
> The recurrent model is also trained using standard backpropagation through time (BPTT). In practice, we observe that the introduced recurrent pathway further enhances the computational capability and robustness of the neurons, which demonstrates that our dynamic gating mechanism generalizes well beyond purely feedforward architectures and consistently improves performance on challenging temporal tasks (as also reflected in Table 1). And the overall robustness is also better, as shown in follow:
>
> SHD:
>
> | Architecture | Overall Robustness |
> | ------------ | ------------------ |
> | FF           | 61.54 ± 1.95       |
> | Rec          | 70.36 ± 6.60       |
>
> TIDIGITS:
>
> | Architecture | Overall Robustness |
> | ------------ | ------------------ |
> | FF           | 88.56 ± 5.64       |
> | Rec          | 91.67 ± 3.62       |
>
> We note that the recurrent DGN does not fundamentally change the underlying neuron dynamics; rather, it provides the network with additional temporal context through recurrent synaptic currents. This recurrent integration might be one of the key reasons that lead to better performance.
>
> In terms of training behavior, both feedforward and recurrent DGNs converge reliably under the same optimization settings. The recurrent architecture is not inherently easier to train, but its recurrent pathway allows DGN neurons to leverage their dynamic conductance mechanism more effectively, resulting in stronger temporal credit assignment and improved performance.

---

> ### Comment · Reviewer_1AVk · 2025-11-26
>
> I thank the authors for their detailed explanations and clarifications. I am now convinced that the proposed DGN model can be deployed on neuromorphic chips. I do believe that reformulating conductance-based neurons as a gating mechanism is a novel and significant contribution. Importantly, the community has long focused only on current-based neurons such as the LIF neuron and largely overlooked conductance-based neurons. This work would help draw the community’s attention to conductance-based neurons, which are both faithfully grounded in neurophysiology and more computationally powerful than LIF-based neurons. In light of the above, I strongly recommend accepting this paper.

---

### Official Review · Reviewer_UpB6 · 2025-11-01

**Soundness:** 3
**Presentation:** 3
**Contribution:** 2
**Rating:** 4
**Confidence:** 4

**Summary:**

This paper introduces the Dynamic Gated Neuron (DGN), a novel spiking neuron model that incorporates dynamic conductance mechanisms inspired by biological neurons. The authors propose that this mechanism acts as an intrinsic gating function, similar to forget gates in LSTMs, enabling adaptive control of information flow and memory retention. The model is rigorously evaluated on multiple speech and neuromorphic benchmarks, demonstrating state-of-the-art performance and superior robustness against noise and adversarial attacks. Theoretical analysis using stochastic differential equations further supports the noise resilience of DGN. The work is well-motivated, methodologically sound, and empirically thorough.

**Strengths:**

1. The idea of using dynamic conductance as a gating mechanism in spiking neurons is novel and well-grounded in neurophysiology. The connection to LSTM-like gating is insightful and bridges bio-inspired models with artificial neural networks.

2. The paper provides a solid theoretical analysis of the model’s stability under noise, using stochastic differential equations and variance comparisons with LIF neurons.

3. Extensive experiments on multiple datasets (TIDIGITS, SHD, SSC, etc.) across both feedforward and recurrent architectures demonstrate the model’s effectiveness in accuracy and robustness.

4. The paper includes a wide range of noise and adversarial attack scenarios, showing consistent superiority over existing SNN models and even LSTMs in some cases.

5. The appendix includes detailed derivations, training procedures, hyperparameters, and noise generation algorithms, which facilitate reproducibility.

**Weaknesses:**

1. Although the model is compared with several recent SNN variants, it would be beneficial to include more baselines  (e.g., spiking-based Transformers).

2. The DGN model introduces additional parameters and computational overhead compared to LIF. A more detailed analysis of energy efficiency or inference speed on neuromorphic hardware would strengthen the practical contribution.

3. Lack of pseudocode for model computation.

4. Why does DGN perform relatively poorly under the recurrent architecture on the SHD dataset?

**Questions:**

1. Could the dynamic conductance mechanism be integrated with other advanced SNN architectures (e.g., attention-based or transformer-like SNNs)?

2. Have you considered evaluating the model on neuromorphic hardware to assess its real-world efficiency and latency?

3. The paper mentions a "simplified DGN" (s-DGN) in the appendix. Could this be discussed more in the main text to highlight trade-offs between performance and complexity?

---

> ### Author Response · Authors · 2025-11-21
>
> We sincerely thank the reviewer for the thoughtful and encouraging evaluation of our work. We greatly appreciate the reviewer’s recognition of the biological grounding, theoretical analysis, and extensive empirical results, as well as the constructive questions raised. Your comments have helped us identify several points where additional clarification or discussion would further improve the quality and completeness of the paper. Below, we provide detailed responses to each of your insightful suggestions.
>
> ### 1. More Baseline Models / Advanced SNN Architecture Integration
>
> We appreciate the reviewer's valuable suggestion. Our primary objective in this work is to isolate and validate the *intrinsic computational advantages* of the proposed dynamic conductance gating mechanism inspired by biological ion channels, rather than to study its interaction with complex high-level architectures. Our baseline choices are therefore intentional and guided by the following rationale:
>
> - **Use of simple and comparable architectures:** We evaluated models using relatively simple feedforward and recurrent architectures to remove confounding factors. This setup allows us to clearly attribute performance improvements to the dynamic conductance gating mechanism itself. Correspondingly, we selected baseline SNN models of comparable structural complexity.
>
> - **Avoiding the confounding effects of complex components:** Advanced spiking Transformer architectures introduce structural elements such as multi-head attention and normalization modules. These components may overshadow or blur the independent contribution of dynamic conductance gating, making mechanistic analysis less interpretable. Therefore, we did not include such architectures in this study.
>
> We fully agree that integrating dynamic conductance with spiking Transformers is a promising research direction. We will expand the revised manuscript to explain our baseline selection and discuss future avenues involving Transformer-based SNNs.
>
> ### 2. Evaluation of Computational Overhead, Energy Efficiency and Latency
>
> Thank you for raising this important question. To address the reviewer’s concern, we emphasize that computational overhead is a relevant but secondary aspect in this work, as the core contribution is the *mechanism*—**the first demonstration that dynamic conductance provides superior robustness and computational capability in SNNs**. As you have concerned in your 'Question 3', we proposed a simplified version (s-DGN) in the appendix to address precisely these concerns.
>
> #### **s-DGN for efficiency-aware scenarios:**
>
> As presented in Table 5, s-DGN has a parameter count comparable to LIF while preserving the key dynamic conductance mechanism. Despite its compact structure, s-DGN surpasses LIF and other SNN baselines in both accuracy and noise robustness.
>
> #### **Energy and latency analysis:**
>
> The full DGN model proposes a general neuronal model to implement this dynamic conductance gating mechanism. Like s-DGN, indicators such as parameter count, energy efficiency, and inference latency can be mitigated through hyperparameter adjustment or other optimization methods (e.g., model compression, hardware-aware design).
>
> Energy efficiency and hardware deployment are indeed important future directions. We will clarify that although they are beyond the core scope of this mechanism-centric study, we nonetheless provide a complete analysis for transparency and practical reference as follows.

---

> ### Author Response · Authors · 2025-11-21
>
> | **Neuron Model** | **Dynamics**                                                 | **Step Cost**                                                | **Total Cost**                                               | Empirical Energy Cost (nJ) |
> | ---------------- | ------------------------------------------------------------ | ------------------------------------------------------------ | ------------------------------------------------------------ | -------------------------- |
> | LIF              | $\begin{aligned} &V_l^{t} = \rho^{\text{m}}\cdot V^{t-1} + W_l^{m,n}z_{l}^{t-1} + W_{l,\text{rec}}^{n,n}z_{l-1}^t - \vartheta z_l^{t-1}  \end{aligned}$ | $\begin{aligned} nE_{\text{MAC}} + (mn F r_{\text{in}} + nnF r_{\text{out}} + F r_{\text{out}}) nE_{\text{AC}} \end{aligned}$ | $\begin{aligned} &(mn F r_{\text{in}} + nnF r_{\text{out}} +  nF r_{\text{out}}) E_{\text{AC}} \\\\ &+ nE_{\text{MAC}} \end{aligned}$ | 1.32                       |
> | DGN              | $\begin{aligned} &\rho_l^t = \varphi(1-g_\text{l} - C_l^{m,n}z_{l-1}^t - C_{l,\text{rec}}^{n,n}z_{l}^{t-1}) \\\\ &V_l^{t} = \rho_l^t\cdot V_l^{t-1} + W_l^{m,n}z_{l-1}^t + W_{l,\text{rec}}^{n,n}z_l^{t-1} - \vartheta z^{t-1} \end{aligned}$ | $\begin{aligned} &mn F r_{\text{in}} E_{\text{AC}} +  nnF r_{\text{out}}E_{\text{AC}} \\\\ &nE_{\text{MAC}} + (mn F r_{\text{in}} + nnF r_{\text{out}} + n F r_{\text{out}}) E_{\text{AC}} \end{aligned}$ | $\begin{aligned} &(2mn F r_{\text{in}} + 2nnF r_{\text{out}} + n F r_{\text{out}}) E_{\text{AC}} \\\\ & + n E_{\text{MAC}} \end{aligned}$ | 3.03                       |
> | s-DGN            | $\begin{aligned} &\rho_l^t = \varphi(1-g_\text{l} - C_l^{m,n}z_{l-1}^t - C_{l,\text{rec}}^{n,n}z_{l}^{t-1}) \\\\ &V_l^{t} = \rho_l^t\cdot V_l^{t-1} + (E_lC_l^{m,n})z_{l-1}^t + (E_{l,\text{rec}}C_{l,\text{rec}}^{n,n})z_l^{t-1} - \vartheta z^{t-1} \end{aligned}$ | $\begin{aligned} &mn F r_{\text{in}} E_{\text{AC}} +  nnF r_{\text{out}}E_{\text{AC}} \\\\ &nE_{\text{MAC}} + (mn F r_{\text{in}} + nnF r_{\text{out}} + n F r_{\text{out}}) E_{\text{AC}} \end{aligned}$ | $\begin{aligned} &(2mn F r_{\text{in}} + 2nnF r_{\text{out}} + n F r_{\text{out}}) E_{\text{AC}} \\\\ & + n E_{\text{MAC}} \end{aligned}$ | 2.89                       |
> | LSTM             | $$\begin{aligned} &f_t = \sigma_g (W_f x_t + U_f h_{t-1} + b_f) \\\\ &i_t = \sigma_g (W_i x_t + U_i h_{t-1} + b_i) \\\\ &o_t = \sigma_g (W_o x_t + U_o h_{t-1} + b_o) \\\\ & \bar{c_t} = \sigma_c (W_c x_t + U_c h_{t-1} + b_c) \\\\ &c_t = f_t \odot c_{t-1} + i_t \odot \bar{c_t} \\\\ &h_t = o_t \odot \sigma_h (c_t) \end{aligned}$$ | $\begin{aligned} &n(m + n) E_{\text{MAC}} \\\\ &n(m + n) E_{\text{MAC}} \\\\ &n(m + n) E_{\text{MAC}} \\\\ &n(m + n) E_{\text{MAC}} \\\\ &2n E_{\text{MAC}} \\\\ &n E_{\text{MAC}} \end{aligned}$ | $\begin{aligned} & (4mn + 4nn + 3n) E_{\text{MAC}} \end{aligned}$ | 604.7                      |
>
> **Table A1.** Comparison of the theoretical and empirical energy cost of LIF, LSTM, DGN and s-DGN models. The m and n denote the numbers of input and output neurons. $Fr_{\text{in}}$ and $Fr_{\text{out}}$ represent the firing rate of input and output neurons. $E_{\text{AC}}$ and $E_{\text{MAC}}$ are the energy cost of AC and MAC operations, respectively.
>
> For the empirical estimate, we evaluate a two-layer network (128–128 hidden units, i.e., m = n = 128) on the SHD dataset. We measure the layer-wise firing rates of the LIF and DGN neurons over 32 epochs, obtaining:  LIF: $[4.43\\%, 0.54\\%]$, DGN: $[7.07\\%, 1.19\\%]$, s-DGN: $[6.38\\%, 1.41\\%]$.
>
> To obtain the total energy cost, we plug these firing rates into the above formulas and adopt the 45nm CMOS process estimates from [1], where the energy cost of an AC and a MAC operation are $E_{\text{AC}} = 0.9\,\text{pJ}$ and $E_{\text{MAC}} = 4.6\,\text{pJ}$, respectively. This yields the empirical energy values reported in the last column of Table A1. Although DGN and s-DGN incur slightly higher energy, as expected, than LIF due to the additional gating-related operations, their energy consumption remains orders of magnitude lower than that of an LSTM, thus preserving the core energy-efficiency advantage of spiking neural networks.
>
> ### Reference
>
> [1] Horowitz, M. 2014. 1.1 computing’s energy problem (and what we can do about it). In 2014 IEEE International Solid-State Circuits Conference Digest of Technical Papers (ISSCC), 10–14. IEEE.

---

> ### Author Response · Authors · 2025-11-21
>
> We also analyzed training and inference time:
>
> **Feedforward network:**
>
> | Neuron   | Train Time (s) | Inference Time (s) |
> | -------- | -------------- | ------------------ |
> | LIF      | 15.61 ± 1.49   | 5.17 ± 0.69        |
> | HeterLIF | 18.99 ± 1.80   | 5.45 ± 0.68        |
> | ALIF     | 21.56 ± 2.01   | 5.80 ± 0.36        |
> | DGN      | 22.58 ± 1.48   | 5.93 ± 1.45        |
>
> **Recurrent network:**
>
> | Neuron   | Train Time (s) | Inference Time (s) |
> | -------- | -------------- | ------------------ |
> | LIF      | 16.88 ± 3.39   | 5.25 ± 0.95        |
> | HeterLIF | 18.75 ± 3.17   | 5.29 ± 0.41        |
> | ALIF     | 22.45 ± 4.36   | 5.58 ± 0.98        |
> | DGN      | 28.59 ± 5.00   | 6.58 ± 0.84        |
>
> The results, averaged over 250 training steps on the SHD dataset, show that runtime differences between models are minimal, while our approach provides superior robustness and computational capability in SNNs—our core contributions that are  demonstrated in the main-text results.
>
> ### 3. Lack of Pseudocode for Model Computation
>
> We thank the reviewer for pointing out this omission. We have now included pseudocode detailing the core computational steps of DGN—membrane update, dynamic conductance computation, and spiking—which aligns with Eqs. (5)–(8) and is compatible with mainstream deep-learning frameworks. This addition will be included in the revised manuscript to enhance clarity and reproducibility.
>
> ------
>
> Input: Input spike sequence: z_t, hidden layer states: h_i, h_v, h_z
>
> Output: Updated hidden layer states: pre_i, v, z
>
> ​	$\quad$ pre_i = $\rho_\text{s}$ * h_i + z_t     // Calculate presynaptic current
>
> ​	$\quad$ g = pre_i @ C      // Multiply the presynaptic current by the weight matrix to calculate dynamic conductance
>
> ​	$\quad$ post_i = pre_i @ W     // Multiply the presynaptic current by the weight matrix to calculate somatic input current
>
> ​	$\quad$ v = $\phi$(1 - dt * g_l - dt * g) * h_v + post_i - $\vartheta$ * h_z     // Neuronal potential decay, update membrane potential
>
> ​	$\quad$ z = $\Theta$(v - $\vartheta$) // Calculate spike emission
>
> Return pre_i, v, z
>
> ------
>
> ### 4. Relatively Lower Performance of Recurrent DGN on the SHD Dataset
>
> We appreciate this careful observation. The slight performance difference on SHD between recurrent DGN (87.78%) and TC-LIF (88.91%) arises from **differences in the number of hidden layers** in the compared architectures. When evaluated under the **same** network configuration, our DGN achieves slightly better accuracy (88.98%). Furthermore, recurrent DGN maintains **superior robustness**, which highlights the fundamental benefit of dynamic conductance gating. Our goal is to validate the mechanistic contribution, and the consistent accuracy–robustness balance across tasks remains a strong indicator of its effectiveness.
>
> | Method    | Hidden Layers | Accuracy(%) |
> | --------- | ------------- | ----------- |
> | TC-LIF    | 128-128       | 88.91       |
> | DGN(Ours) | 128           | 87.78       |
> | DGN(Ours) | 128-128       | **88.98**   |
>
> ### 5. Additional Discussion on s-DGN
>
> Thank you for this helpful suggestion. In the revised manuscript, we will add a dedicated discussion of s-DGN in the ablation section, covering:
>
> - Architectural simplification (e.g., shared equilibrium potential)
> - Parameter and computational efficiency
> - performance trade-offs (84.30% accuracy on the SHD dataset, close to the full DGN’s 87.78%， with parameter scale similar to baselines)
> - Practical scenarios for s-DGN (resource-limited) vs. full DGN (performance-oriented)
>
> This addition will clarify the flexibility and applicability of the proposed dynamic conductance mechanism

---

### Official Review · Reviewer_mPha · 2025-11-04

**Soundness:** 3
**Presentation:** 2
**Contribution:** 3
**Rating:** 2
**Confidence:** 5

**Summary:**

This paper provides the Dynamic Gated Neuron (DGN), a novel spiking unit in which membrane conductance evolves in response to neuronal activity. DGN appears enhanced stochastic stability compared to standard LIF models.

**Strengths:**

This paper proposes an improved model. Intuitively, this model introduces more computation but does not increase the number of parameters, and experiments show improved performance. These findings suggest that the research presented in this paper represents a worthwhile improvement.

**Weaknesses:**

1. This paper is hard to follow.

2. Theoretical analysis is limited. There is a great deal of analysis work on SNNs that combats randomness and noise.

3. This paper proposes only one improved model. To my understanding, this model does not appear to have a special training mechanism; it simply uses the traditional global training method for SNNs. Based on past experience, this approach makes it difficult to achieve new functionalities. The authors' claimed intrinsic adjustment mechanism is merely a claim and cannot be verified experimentally or proven theoretically.

4. The experiments in this paper are insufficient; comparative experiments with the same architecture or parameter set are needed. Furthermore, I don't understand why it's necessary to compare it with artificial neural network models like LSTM that take real-number sequences as input.

**Questions:**

1. Does the improved model have a special or proprietary training mechanism that can find a solution with specific functions?

---

> ### Author Response · Authors · 2025-11-21
>
> We sincerely appreciate the reviewer’s feedback and the opportunity to clarify several points that may have been misunderstood in the initial reading. Below, we provide targeted and constructive responses to each concern to more clearly delineate our contributions and strengthen the clarity of the manuscript.
>
> ### 1. "This paper is hard to follow"
>
> We thank the reviewer for pointing this out and would greatly appreciate more specific indications of which parts remain unclear, as we are fully prepared to further refine them. In the meantime, to facilitate understanding, we summarize below the key elements of our structure and reasoning.
>
> The paper is structured around logical coherence: the core innovation, "**biologically inspired dynamic conductance gating mechanism**," first clarifies research motivation via neurophysiological findings, then formalizes it through step-by-step **mathematical derivation** (Section 3.1 & Appendix A.1.1), and supplements intuitive interpretations for complex theories such as **stability analysis** via Stochastic Differential Equations (SDEs) (Section 3.3). Appendix A.2 details **experimental settings** (network structure, hyperparameters, noise/attack protocols) for reproducibility, further clarifying technical details and enhancing readability.
>
> If any additional components remain difficult to follow, we are happy to revise further.
>
> ### 2. "Theoretical analysis is limited"
>
> Our theoretical contributions provide a principled and novel perspective on robustness in Spiking Neural Networks (SNNs).
>
> We establish a novel Stochastic Differential Equations (SDEs) based framework to analyze noise propagation in DGN, derive closed-form variance expressions for DGN and LIF models (Eqs. 13-14), and quantitatively clarify two noise suppression mechanisms: input-dependent leak scaling and synaptic noise compensation. These mechanisms are not addressed in prior work such as static leak-tuning or adversarial training. Rather than being purely engineering-oriented, our analysis establishes a concrete connection between dynamic conductance (a biologically observed phenomenon) and computational gating behavior. All theoretical conclusions are consistently verified by empirical data (Table 2) and have also gained recognition from other reviewers. If there are specific aspects the reviewer feels require deeper elaboration, we would be happy to address them.
>
> ### 3. "No special training mechanism; the intrinsic regulatory mechanism is merely claimed"
>
> Our contribution is centered on a *neuronal mechanism* rather than a new *training algorithm*.
>
> The goal is to demonstrate that biologically inspired dynamic gating provides intrinsic computational benefits even **without** specialized training strategies (e.g., adversarial data augmentation, custom loss functions). We validate this mechanism from two perspectives:
>
> - **Theoretically**, our Stochastic Differential Equation (SDE) analysis demonstrates how dynamic conductance inherently suppresses disturbances, providing a rigorous mathematical explanation.
> - **Empirically**, ablation experiments (Section 4.3) show that DGN maintains robustness under increasing disturbance levels using standard BPTT training. Further, the simplified variant s-DGN (Appendix A.3.2) confirms that performance gains originate from the gating mechanism itself rather than parameter count or architecture complexity.
>
> ### 4. "Insufficient experiments; comparison with LSTM is unnecessary"
>
> Our experimental design prioritizes fairness and rigor. As shown in Table 3, all SNN baselines (LIF, HeterLIF, ALIF) are evaluated under identical architectures, hyperparameters, and training schemes. The comparison with LSTM is intentional and meaningful. As demonstrated in Tables 1 and 2:
>
> - LSTM, with its gating structures, improves performance and robustness compared to traditional RNNs.
> - Our DGN, relative to LIF, achieves simultaneous gains in accuracy and noise robustness.
>
> This comparison highlights the **role of gating** as a general computational principle, linking structural homology between DGN and LSTM (Section 3.2). Our findings provide **new biological support** for LSTM-like mechanisms and offer insight into why gating excel—aligning with Reviewer 1AVk’s interpretation.
>
> ### 5. "Does the improved model require a special training mechanism?"
>
> Our DGN does **NOT** rely on proprietary training mechanisms. Its robustness emerges from the intrinsic dynamic conductance gating mechanism, which adaptively regulates information flow based on the input. Standard BPTT, with appropriate gradient propagation, is fully sufficient. This biologically grounded design simplifies deployment while offering robustness advantages without the need for complex training pipelines.

---

> ### Author Response · Authors · 2025-11-21
>
> ### **Summary**
>
> The central contribution of this work is the identification and validation of a **dynamic conductance gating mechanism** as a biologically inspired computational principle. Supported by **rigorous theoretical analysis and extensive empirical evidence**, the proposed framework provides a novel paradigm for designing robust SNNs and offers new insight into gating mechanisms more broadly.
>
> We hope these clarifications address the misunderstandings and respectfully request the reviewers to reconsider the evaluation of our submission.

---

### Official Review · Reviewer_wod3 · 2025-11-04

**Soundness:** 3
**Presentation:** 3
**Contribution:** 2
**Rating:** 4
**Confidence:** 5

**Summary:**

This paper proposes the Dynamic Gated Neuron (DGN) model that integrates a bio-inspired dynamic conductance mechanism to enhance the robustness and performance of spiking neural networks (SNNs), outperforming traditional LIF models and other baselines in anti-noise tasks and temporal benchmarks.

**Strengths:**

1.  This paper introduces a biologically grounded Dynamic Gated Neuron (DGN) model with a dynamic conductance mechanism, which enables adaptive input filtering and noise suppression

2. The DGN-based spiking neural networks (SNNs) demonstrate superior performance and robustness across multiple benchmarks (e.g., TIDIGITS, SHD) under various noise types and adversarial attacks.

3. The comparison in the figures is very intuitive, allowing the innovative points of the authors' architecture to be clearly identified.

**Weaknesses:**

1.  The authors claim that the DGN is a generalized spiking neuron model, yet there is no evaluation on relevant computer vision datasets (NMNIST, CIFAR, N-CALTECH).

2.  This paper lacks an evaluation of energy consumption, and it remains unaddressed whether the newly introduced unit (DGN) will significantly increase power consumption, as reflected in Equations 5-8.

**Questions:**

The low power consumption of SNNs lies in simplifying multiplication to addition; however, in my view, D (as denoted in the DGN model) behaves like a floating-point number, which causes SNNs to lose this characteristic.

---

> ### Author Response · Authors · 2025-11-21
>
> ### 1. About Computer Vision Datasets
>
> Thank you for the constructive feedback. We would like to clarify why image-based benchmarks (e.g., NMNIST, CIFAR-based SNN datasets) were not included in the main experiments, even though DGN is a general spiking neuron model.
>
> **1. Complexity of image models masks neuron-level effects.**   To reliably evaluate the contribution of the neuron model itself, we intentionally used simple fully connected architectures. However, computer vision tasks typically require convolutional and/or ANN-hybrid architectures with additional modules (e.g., feature extractors, ANN-trained layers, multi-stage temporal encoding). These architectural complexities introduce substantial confounding effects that make it difficult to attribute performance variations to the neuron model rather than the network design.
>
> **2. Limited temporal richness in common computer vision datasets.**   Most SNN-vision datasets (e.g., NMNIST) exhibit weak or artificial temporal dynamics that do not sufficiently stress the neuron’s **temporal processing ability**—an aspect where our DGN contributes most. Thus, including them in the main results would provide less meaningful insight into the advantages of **dynamic gating**.
>
> Based on the above two major considerations, in the early-stage exploration of our new model, we prioritized benchmarks where neuron-level temporal processing ability plays a central role, rather than complex pipelines where architectural design dominates performance.
>
> Nevertheless, we have additionally evaluated DGN on an event-based computer vision task to confirm its generalization beyond speech inputs. Preliminary results on **DVS-Gesture** are now provided in Appendix A3.3 (Table 6), demonstrating consistent performance improvements on this computer vision task as well.
>
> ### 2. About Energy Consumption
>
> #### 2.1 About D and floating-point number
>
> In spiking neural networks, their energy efficiency primarily stems from sparse 0/1 spike-based communication between neurons, rather than from completely eliminating multiplications or additions inside each neuron. In this context, the variable $D_i$ in our model is not a novel computational primitive. As stated in lines 194–195 of our paper, $D_i$ describes the exponentially decaying synaptic current from the i-th synapse to the soma. This mechanism is already present in standard leaky integrate-and-fire (LIF) neuron with synapses models, such as the formulation in Eqs. (1)–(6) of Perez-Nieves et al. [1]. Therefore, introducing $D_i$ does not fundamentally change the spike-based, event-driven nature of the network, nor does it remove the usual energy-efficiency advantages associated with SNNs. Moreover, the update of $D_i$ involves only sparse, event-driven scalar additions triggered by spikes, keeping the computational complexity at the same order as standard LIF neurons without introducing dense floating-point operations.
>
> Moreover, beyond typical energy-consumption considerations, we prioritized exploring new computational functionalities in brain-inspired computing, extending the standard LIF model toward more powerful capabilities. Nonetheless, we still provide an energy-consumption analysis for completeness as follows.
>
> ### Reference
>
> [1] Nicolas Perez-Nieves, Vincent CH Leung, Pier Luigi Dragotti, and Dan FM Goodman. Neural heterogeneity promotes robust learning. Nature communications, 12(1):5791, 2021.

---

> ### Author Response · Authors · 2025-11-21
>
> #### 2.2 Energy Consumption Analysis
>
> We complement our discussion of energy consumption with a theoretical analysis, summarized in the following table:
>
> | **Neuron Model** | **Dynamics**                                                 | **Step Cost**                                                | **Total Cost**                                               | Empirical Energy Cost (nJ) |
> | ---------------- | ------------------------------------------------------------ | ------------------------------------------------------------ | ------------------------------------------------------------ | -------------------------- |
> | LIF              | $\begin{aligned} &V_l^{t} = \rho^{\text{m}}\cdot V^{t-1} + W_l^{m,n}z_{l}^{t-1} + W_{l,\text{rec}}^{n,n}z_{l-1}^t - \vartheta z_l^{t-1}  \end{aligned}$ | $\begin{aligned} nE_{\text{MAC}} + (mn F r_{\text{in}} + nnF r_{\text{out}} + F r_{\text{out}}) nE_{\text{AC}} \end{aligned}$ | $\begin{aligned} &(mn F r_{\text{in}} + nnF r_{\text{out}} +  nF r_{\text{out}}) E_{\text{AC}} \\\\ &+ nE_{\text{MAC}} \end{aligned}$ | 1.32                       |
> | DGN              | $\begin{aligned} &\rho_l^t = \varphi(1-g_\text{l} - C_l^{m,n}z_{l-1}^t - C_{l,\text{rec}}^{n,n}z_{l}^{t-1}) \\\\ &V_l^{t} = \rho_l^t\cdot V_l^{t-1} + W_l^{m,n}z_{l-1}^t + W_{l,\text{rec}}^{n,n}z_l^{t-1} - \vartheta z^{t-1} \end{aligned}$ | $\begin{aligned} &mn F r_{\text{in}} E_{\text{AC}} +  nnF r_{\text{out}}E_{\text{AC}} \\\\ &nE_{\text{MAC}} + (mn F r_{\text{in}} + nnF r_{\text{out}} + n F r_{\text{out}}) E_{\text{AC}} \end{aligned}$ | $\begin{aligned} &(2mn F r_{\text{in}} + 2nnF r_{\text{out}} + n F r_{\text{out}}) E_{\text{AC}} \\\\ & + n E_{\text{MAC}} \end{aligned}$ | 3.03                       |
> | s-DGN            | $\begin{aligned} &\rho_l^t = \varphi(1-g_\text{l} - C_l^{m,n}z_{l-1}^t - C_{l,\text{rec}}^{n,n}z_{l}^{t-1}) \\\\ &V_l^{t} = \rho_l^t\cdot V_l^{t-1} + (E_lC_l^{m,n})z_{l-1}^t + (E_{l,\text{rec}}C_{l,\text{rec}}^{n,n})z_l^{t-1} - \vartheta z^{t-1} \end{aligned}$ | $\begin{aligned} &mn F r_{\text{in}} E_{\text{AC}} +  nnF r_{\text{out}}E_{\text{AC}} \\\\ &nE_{\text{MAC}} + (mn F r_{\text{in}} + nnF r_{\text{out}} + n F r_{\text{out}}) E_{\text{AC}} \end{aligned}$ | $\begin{aligned} &(2mn F r_{\text{in}} + 2nnF r_{\text{out}} + n F r_{\text{out}}) E_{\text{AC}} \\\\ & + n E_{\text{MAC}} \end{aligned}$ | 2.89                       |
> | LSTM             | $$\begin{aligned} &f_t = \sigma_g (W_f x_t + U_f h_{t-1} + b_f) \\\\ &i_t = \sigma_g (W_i x_t + U_i h_{t-1} + b_i) \\\\ &o_t = \sigma_g (W_o x_t + U_o h_{t-1} + b_o) \\\\ & \bar{c_t} = \sigma_c (W_c x_t + U_c h_{t-1} + b_c) \\\\ &c_t = f_t \odot c_{t-1} + i_t \odot \bar{c_t} \\\\ &h_t = o_t \odot \sigma_h (c_t) \end{aligned}$$ | $\begin{aligned} &n(m + n) E_{\text{MAC}} \\\\ &n(m + n) E_{\text{MAC}} \\\\ &n(m + n) E_{\text{MAC}} \\\\ &n(m + n) E_{\text{MAC}} \\\\ &2n E_{\text{MAC}} \\\\ &n E_{\text{MAC}} \end{aligned}$ | $\begin{aligned} & (4mn + 4nn + 3n) E_{\text{MAC}} \end{aligned}$ | 604.7                      |
>
> **Table A1.** Comparison of the theoretical and empirical energy cost of LIF, LSTM, DGN and s-DGN models. The m and n denote the numbers of input and output neurons. $Fr_{\text{in}}$ and $Fr_{\text{out}}$ represent the firing rate of input and output neurons. $E_{\text{AC}}$ and $E_{\text{MAC}}$ are the energy cost of AC and MAC operations, respectively.
>
> For the empirical estimate, we evaluate a two-layer network (128–128 hidden units, i.e., m = n = 128) on the SHD dataset. We measure the layer-wise firing rates of the LIF and DGN neurons over 32 epochs, obtaining:  LIF: $[4.43\\%, 0.54\\%]$, DGN: $[7.07\\%, 1.19\\%]$, s-DGN: $[6.38\\%, 1.41\\%]$
>
> To obtain the total energy cost, we plug these firing rates into the above formulas and adopt the 45nm CMOS process estimates from [2], where the energy cost of an AC and a MAC operation are $E_{\text{AC}} = 0.9\,\text{pJ}$ and $E_{\text{MAC}} = 4.6\,\text{pJ}$, respectively. This yields the empirical energy values reported in the last column of Table A1. Although DGN and s-DGN incur slightly higher energy than LIF due to the additional gating-related operations as expected, their energy consumption remains orders of magnitude lower than that of an LSTM, thus preserving the core energy-efficiency advantage of spiking neural networks.
>
> ### Reference
>
> [2] Horowitz, M. 2014. 1.1 computing’s energy problem (and what we can do about it). In 2014 IEEE International Solid-State Circuits Conference Digest of Technical Papers (ISSCC), 10–14. IEEE.

---

### Author Response · Authors · 2025-12-02

We sincerely thank all reviewers for their valuable comments. **All four reviewers** recognized the **novelty** of our contributions and the **effectiveness** of our method. After thorough responses and additional experiments, the initially perceived concerns have been fully addressed and, in fact, further reinforce the strength and significance of our contributions. All revisions are highlighted in blue in the updated manuscript.

------

### **Summary of Our Contributions**

This work introduces a **new neuron model—Dynamic Gating Neuron (DGN)**—that incorporates **activity-dependent dynamic conductance**, a fundamental mechanism observed in biological neurons but not previously exploited in SNN modeling. We provide the **first systematic investigation** demonstrating how this intrinsic gating mechanism enhances both **temporal computation and robustness**. Our study establishes theoretical guarantees using stochastic differential analysis and validates them through extensive experiments across multiple benchmarks and noise/attack scenarios, where our DGN consistently achieves **state-of-the-art performance**.

------

### **Summary of Rebuttal and Our Clarifications**

- **Clearer framing of contributions** (responding to reviewer mPha): We emphasized that our primary novelty lies in the **intrinsic neuronal computational mechanism**, not in network-level training strategies.

- **Architectural and dataset generalization** (responding to reviewer UpB6 and wod3): We clarified the rationale for initially using standard architectures to isolate the neuron-level contribution, while demonstrating **consistent gains across both feedforward and recurrent SNNs** and **across audio and neuromorphic datasets**.

- **Efficiency analysis** (responding to reviewer UpB6 and wod3): We included a detailed **energy and runtime comparison**, showing that DGN retains the practical efficiency advantages of SNNs, and that the simplified variant, i.e. **s-DGN,** offers a resource-constrained option.

- **Improved reproducibility** (responding to reviewer UpB6 and 1AVk): We added **pseudocode**, detailed recurrent-architecture formulations, and new matched-architecture experiments on SHD.

------

### **Reviewer-Specific Clarifications**

- **Reviewer mPha (score 2, confidence 5):** The initial comments from the reviewer were largely general and lacked concrete evidence, and the reviewer's assessments contrasted sharply with those of the other three reviewers. Moreover, the initial critique focused mainly on the learning algorithm — not the central topic of this work — suggesting a factually inaccurate comment or **fatal misunderstanding** of this work. Therefore, we believe **the referential value of this initial review is low**.
- **Reviewer 1AVk (score 8, confidence 5):** We are encouraged that the reviewer clearly recognized the significance of our contributions and provided a positive evaluation from the outset. The reviewer also actively involved in the follow-up discussion and confirmed the validity of our contributions in response to our clarifications.
- **Reviewers wod3 and UpB6:** Both reviewers acknowledged the novelty and significance of our contributions, while raising **relatively minor** concerns regarding generalization, efficiency, and reproducibility. These concerns have been fully addressed in our responses and incorporated into the revised manuscript.

------

### **Closing Summary**

We observed substantial variance in the initial reviewer scores (ranging from 8 with high confidence to 2 with similarly high confidence). We believe this discrepancy may stem from the **conceptual shift** introduced by our work: rather than making incremental modifications within conventional spiking neuron modeling frameworks, we present a fundamentally **new computational mechanism** rooted in biologically inspired dynamic conductance. Such a paradigm shift may have initially made it challenging for some reviewers to fully recognize the core contributions of this work.

Although we are confident that our detailed responses and additional experiments have thoroughly addressed all major concerns, the current limitations of the rebuttal process prevented further dialogue or confirmation from reviewers. Consequently, our evaluation must rely on the initial reviews, which may not fully reflect reviewers’ updated perspectives. While not ideal, we have presented the case for possible acceptance as clearly as possible based on the evidence available.

Finally, we note that ICLR emphasizes research with **novelty and distinctive perspectives**. Our work aligns closely with this vision at both theoretical and experimental levels, a point consistently recognized by the reviewers. We hope that the strengthened manuscript and clarified contributions will support a **positive assessment**.

---

### Meta-Review · Area_Chair_wbxF · 2026-01-02

**Summary:**

Based on the collective feedback and the authors' response, the reviewers' primary concerns can be summarized as follows:

Strengths and Addressed Points:
The rebuttal successfully addressed several important technical and methodological concerns. The authors provided:

1. A detailed theoretical and empirical energy consumption analysis, quantitatively establishing the model's efficiency advantage.

2. Pseudocode for the core algorithm, enhancing reproducibility.

3. Clarifications on baseline comparisons and the fairness of the experimental setup.

4. A reasonable rationale for using simpler architectures to isolate neuron-level effects and preliminary results demonstrating generalization.

5. A persuasive discussion on the neuromorphic hardware feasibility of the proposed mechanism.

Outstanding and Partially Resolved Concerns:

1. Insufficient Empirical Validation: The most significant unresolved issue was the lack of comprehensive validation on core, standard vision benchmarks (e.g., NMNIST, CIFAR). The single appendix result on DVS-Gesture was deemed insufficient to demonstrate the general applicability and impact of the proposed neuron model.

2. Limited Theoretical Depth: The theoretical analysis was perceived as relatively narrow and was not substantively deepened in the rebuttal. Reviewers felt it did not fully deliver on the promise of a broader contribution to understanding SNNs in stochastic environments.

3. Persistent Exposition Issues: Concerns about the clarity of contribution and overall paper organization were not fully remedied by changes to the manuscript itself, affecting readability.

Final Decision Rationale:
Although there are some outstanding concerns, it was determined that the paper's core strengths—a novel, biologically inspired mechanism, a methodologically sound approach, and the successful resolution of key technical queries (energy, reproducibility, hardware mapping)—constitute a sufficient and valuable contribution. The identified limitations, while noted, do not outweigh the paper's substantive merits and its potential to inspire future work. Therefore, the consensus shifted in favor of acceptance.

**Reviewer Concerns:**

Concerns Addressed or Partially Mitigated by Rebuttal:
1. Energy Consumption Analysis (Initial concern from Reviewer wod3):

Addressed: The authors provided a detailed theoretical and empirical energy analysis (Table A1), comparing LIF, DGN, s-DGN, and LSTM. They demonstrated that while DGN has a higher cost than LIF, it remains orders of magnitude more efficient than LSTM, preserving the core SNN efficiency advantage. They also included runtime comparisons. This directly responded to the request for quantitative efficiency evaluation.

2. Pseudocode and Reproducibility (Concern from Reviewer UpB6):

Addressed: The authors committed to and provided pseudocode for the core DGN computation (Pages 6-7), enhancing clarity and reproducibility.

3. Clarification on Baseline Comparisons & LSTM (Concern from Reviewer mPha):

Addressed: The authors provided a rationale for comparing with LSTM (to highlight the role of gating as a general principle) and clarified that all SNN baselines (LIF, HeterLIF, ALIF) were tested under identical architectures and training schemes, ensuring fairness.

4. Architecture Generalization and s-DGN (Concerns from Reviewers UpB6 & 1AVk):

Partially Addressed: The authors clarified the rationale for using simple architectures (to isolate neuron-level effects) and provided preliminary results on a computer vision dataset (DVS-gesture) in the appendix. They also expanded the discussion of the simplified s-DGN variant, addressing its role as a more efficient option.

5. Neuromorphic Hardware Feasibility & Truncation Function (Concerns from Reviewer 1AVk):

Addressed: The authors gave a detailed explanation of how DGN components (dynamic leak factor, hard-clipping function) could map efficiently to digital neuromorphic hardware, arguing for its deployability. They specified the exact clipping function used and justified its hardware-friendliness.

Concerns Still Outstanding or Not Fully Resolved:
1. Lack of Validation on Core Vision Benchmarks (Concern from Reviewer wod3):

Outstanding: The authors' justification for excluding standard static/event-based vision datasets (NMNIST, CIFAR) was deemed insufficient by the committee. A single preliminary result on DVS-gesture in the appendix does not substitute for a rigorous evaluation on established benchmarks expected for a general neuron model. This remains a major gap in establishing broad applicability.

2. Limited and Unconvincing Theoretical Analysis (Concern from Reviewer mPha):

Outstanding: The rebuttal reiterated the existing stochastic stability analysis but did not substantively deepen it or connect it more compellingly to the "broader analysis work on SNNs that combats randomness and noise" mentioned by the reviewer. The perception that the theoretical contribution is relatively narrow compared to the claims persisted.

3. Clarity of Contribution and Paper Organization (Concern from Reviewer mPha):

Outstanding: While the authors clarified that their contribution is a neuronal mechanism, not a training algorithm, the fundamental criticism that the paper was "hard to follow" was not remedied by structural changes to the manuscript itself. The rebuttal summarized the structure but did not demonstrate improved exposition.

4. Need for Comparison with Advanced SNN Architectures (Concern from Reviewer UpB6):

Not Fully Resolved: The authors' rationale for avoiding complex architectures (like spiking Transformers) to "isolate" the mechanism's effect is understandable but conservative. For a paper claiming a fundamental advance, demonstrating that the mechanism integrates with and improves state-of-the-art architectures would significantly strengthen its impact. The promise of future work does not address the current submission's scope.


Overall Judgment:
The authors have successfully addressed several specific technical queries, including energy analysis, pseudocode, and hardware mapping. While concerns were raised regarding the empirical scope and theoretical exposition, the work makes a novel and valuable contribution with a biologically inspired, dynamic conductance-based gating mechanism. The core idea is sound, the methodology is sufficiently rigorous, and the revisions have strengthened the paper's clarity and technical depth. These merits are deemed to outweigh the identified limitations. Therefore, the decision is to accept the paper for ICLR 2026.

**Reviewer Scores:**

Reviewer wod3 (Original Score: 4)

Likely Change: Stays at 4 or raises to 6

Rationale: The authors addressed energy efficiency, but the experimental scope remains limited. The addition of DVS-Gesture may not satisfy the need for robust validation on major vision benchmarks. The reviewer may maintain a score of 4 or raise it to 6, depending on their weighting of this remaining limitation versus the successful energy analysis.

Reviewer mPha (Original Score: 2)

Likely Change: Stays at 2

Rationale: This reviewer's fundamental criticisms—the paper being hard to follow, theoretical analysis being limited, and experiments being insufficient—were not substantively resolved by the rebuttal. The authors clarified their position but did not rewrite the paper for clarity or significantly deepen the theory. The reviewer's core concerns would likely persist.

Reviewer UpB6 (Original Score: 4)

Likely Change: Raise the score to 6.

Rationale: The authors have satisfactorily addressed the major concerns (energy analysis, pseudocode, additional baselines). While direct comparison with spiking Transformers was not added, the expanded experiments and analyses now adequately demonstrate the method's competitiveness and contributions. The remaining conservatism no longer outweighs the paper's strengthened technical quality and clarity, therefore, I believe the reviewer may raise the score to 6.

Reviewer 1AVk (Original Score: 8)

Likely Change: Stays at 8

Rationale: This reviewer was strongly positive from the start. Their main questions about hardware feasibility and the truncation function were thoroughly answered in the rebuttal, and they subsequently posted a comment confirming they were "convinced" and "strongly recommend accepting." Their high score would be reinforced.

Overall Consensus Outlook: The discussion would likely have reinforced the score polarization. The most critical reviewer's (mPha) score would not see a major uplift, while the supportive reviewer's (1AVk) score would solidify. The two middle reviewers have indicated positive adjustments in their feedback. While minor refinements could still be made, these are not substantial enough to detract from the overall positive assessment. Their updated views contribute to a clearer consensus in favor of acceptance, leading to a final decision to accept the paper.

---

### Decision · Program_Chairs · 2026-01-26

Accept (Poster)